# 3D Molecule Generation from Rigid Motifs via $\mathrm{SE}(3)$ Flows

## Abstract

Three-dimensional molecular structure generation is typically performed at the level of individual atoms, yet molecular graph generation techniques often consider *fragments* as their structural units. Building on the advances in *frame-based* protein structure generation, we extend these fragmentation ideas to 3D, treating general molecules as sets of *rigid-body motifs*. Utilising this representation, we employ $\mathrm{SE}(3)$-equivariant generative modelling for *de novo* 3D molecule generation from rigid motifs. In our evaluations, we observe comparable or superior results to state-of-the-art across benchmarks, surpassing it in atom stability on GEOM-DRUGS, while yielding a $2\times$ to $10\times$ reduction in generation steps and offering $3.5\times$ compression in molecular representations compared to the standard atom-based methods.

## 1. Introduction

The generation of 3D molecular structures is a cornerstone of *in silico* drug discovery and material design. Recent advances in deep learning have enabled the development of powerful generative models that treat molecules as point clouds of atoms, utilising E(3)- and SE(3)-equivariant diffusion-based frameworks to determine atomic coordinates (Hoogeboom et al., 2022; Xu et al., 2023). While these atom-based approaches achieve impressive performance, they operate at a low level of abstraction, discarding the rich chemical modularity inherent to molecular structures. In the realm of molecular graph generation, however, this hierarchical nature is widely exploited by fragment-based methods (Jin et al., 2018; Hetzel et al., 2025), which assemble molecules from chemically meaningful motifs or scaffolds to capture high-level semantics and ensure validity. In the domain of 3D protein structure generation, such a modular perspective has also proven highly effective. Semi-

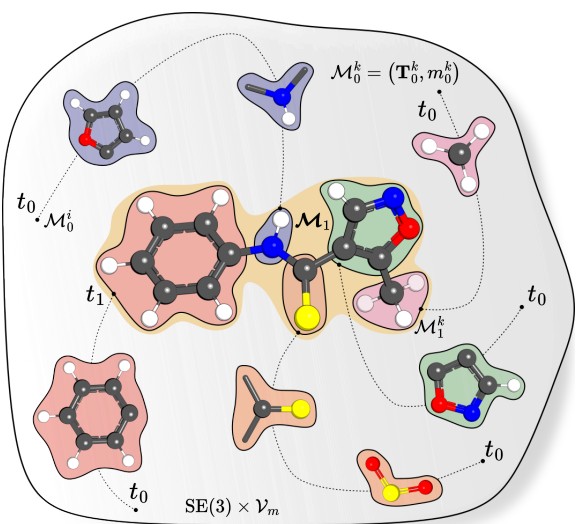

*Figure 1.* Molecule $\mathcal{M}$ is generated from motifs $\mathcal{M}$ in a joint flow on the product space of rigid frames $\mathbf{T}$ and motif classes $m$.

nal works such as ALPHAFOLD2 (Jumper et al., 2021) and FRAMEDIFF (Yim et al., 2023) demonstrated the efficacy of abstracting amino acid residues as *rigid frames* in $\mathrm{SE}(3)$, decoupling backbone geometry from local side-chain details. However, lifting these ideas to general drug-like molecule generation in 3D presents significant challenges: unlike proteins, which are linear chains of a fixed set of residues, small molecules exhibit arbitrary branching, diverse topologies, and a vastly larger vocabulary of chemical motifs.

In this work, we propose to bridge this gap by treating general molecules as collections of *rigid motifs*. By decomposing molecules into chemically meaningful rigid fragments, we lift the generative task from the $\mathbb{R}^3$ atom space to the $\mathrm{SE}(3)$ manifold of fragments. We adopt a multimodal flow matching framework that jointly learns the discrete distribution of motif types and their continuous spatial configuration, illustrated in Figure 1. This formulation enables us to generate high-fidelity molecular structures with significantly fewer sampling steps, leveraging chemically inspired representations that are substantially more concise than those of standard all-atom approaches.

**Contributions**  Our primary contributions are as follows. We propose MOTIFLOW, a framework for 3D molecule generation that parametrises molecules as sets of rigid-body

[1]Anonymous Institution, Anonymous City, Anonymous Region, Anonymous Country. Correspondence to: Anonymous Author <anon.email@domain.com>.

Preliminary work. Under review by the International Conference on Machine Learning (ICML). Do not distribute.

motifs in $\mathrm{SE}(3)$ rather than individual atoms. We develop a data-driven fragmentation and canonicalisation strategy that handles arbitrary molecular topologies and accounts for motif symmetries. Further, we adapt the multimodal flow (Campbell et al., 2024) to the task of *de novo* 3D structure generation of drug-like molecules. This formulation allows us to natively handle the joint generation of discrete fragment identities and their continuous geometric configurations without relying on autoregressive steps or learned decoding back onto the atom-level structure. Finally, we empirically demonstrate that our method achieves superior performance compared to state-of-the-art atom-based baselines on medium- and large-sized molecules in the GEOM-DRUGS benchmark (Axelrod & Gomez-Bombarelli, 2022) and scales to the larger molecules of the QMUGS dataset (Isert et al., 2022). MOTIFLOW produces high-quality molecular structures using an order of magnitude fewer generation steps and excels at conditional generation tasks.

## 2. Background

In this section, we provide a brief overview of the theoretical concepts necessary to establish our generative framework.

### 2.1. Flow Matching on $\mathrm{SE}(3)$ Manifold

**Geometric Preliminaries** The group of rigid motions $\mathrm{SE}(3) \cong \mathbb{R}^3 \rtimes \mathrm{SO}(3)$ describes the configuration of rigid bodies in 3D space. While the translational component $\mathbb{R}^3$ is Euclidean (see Appendix A.1), the rotational one $\mathrm{SO}(3)$ is a *compact Lie group* – a smooth manifold equipped with a group structure. To define flows on this curved space, we rely on Riemannian geometry. At any point $\mathbf{R} \in \mathrm{SO}(3)$, the *tangent space* $\mathcal{T}_{\mathbf{R}}\mathrm{SO}(3)$ is the vector space containing all possible velocity vectors passing through $\mathbf{R}$. We equip the manifold with the canonical bi-invariant *Riemannian metric* defined by the inner product $\langle \mathbf{U}, \mathbf{V} \rangle_{\mathbf{R}} = \frac{1}{2}\mathrm{Tr}\left(\mathbf{U}^\top \mathbf{V}\right)$ for tangent vectors $\mathbf{U}, \mathbf{V} \in \mathcal{T}_{\mathbf{R}}\mathrm{SO}(3)$. This metric induces the norm $\|\mathbf{U}\|_{\mathrm{SO}(3)} = \sqrt{\langle \mathbf{U}, \mathbf{U} \rangle_{\mathbf{R}}}$. The bijective mapping between the tangent space and the manifold is handled by the *exponential map* $\mathbf{Q} = \exp_{\mathbf{R}}(\mathbf{V})$, which projects a tangent vector $\mathbf{V}$ along a geodesic to a point $\mathbf{Q}$ on the manifold, and its inverse, the *logarithmic map* $\mathbf{V} = \log_{\mathbf{R}}(\mathbf{Q})$, which recovers the tangent vector, connecting $\mathbf{R}$ to $\mathbf{Q}$.

$\mathrm{SE}(3)$ **Flow Matching** We decouple the generative process on $\mathrm{SE}(3)$ into independent translational and rotational components (Yim et al., 2023). For the rotational component, we follow the FOLDFLOW-BASE framework of Bose et al. (2023). This method extends *Riemannian flow matching* (Chen & Lipman, 2024) by deriving a closed-form expression for the ground-truth conditional vector field, thereby significantly improving training speed and stability.

We define a probability path $\mathbf{R}_t$ that interpolates along the geodesic connecting a sample from the uniform prior $\mathbf{R}_0 \sim \mathcal{U}_{\mathrm{SO}(3)}$ to a data sample $\mathbf{R}_1$. The conditional vector field

$u_t^{\mathbf{R}}(\mathbf{R}_t \mid \mathbf{R}_1) \in \mathcal{T}_{\mathbf{R}_t}\mathrm{SO}(3)$ generating this path is:

$$u_t^{\mathbf{R}}(\mathbf{R}_t \mid \mathbf{R}_1) = \frac{1}{1-t}\log_{\mathbf{R}_t}(\mathbf{R}_1).$$

Computing $\log_{\mathbf{R}_t}(\mathbf{R}_1)$ naively requires evaluating an infinite matrix power series, which is computationally expensive and numerically unstable. To circumvent this, Bose et al. (2023) exploits the Lie group structure. Instead of computing the logarithm directly at $\mathbf{R}_t$, they first compute the relative rotation $\mathbf{R}_{\mathrm{rel}} = \mathbf{R}_t^\top \mathbf{R}_1$. This maps the problem to the identity-tangent space, the Lie algebra $\mathfrak{so}(3)$, where the logarithm admits a fast, closed-form solution via the Rodrigues' formula. The resulting vector is then transported back to the tangent space at $\mathbf{R}_t$ via left-multiplication, rendering the calculation efficient and exact.

To train the model, one samples intermediate noisy frames $\mathbf{T}_t = (\mathbf{R}_t, \mathbf{x}_t)$. The rotation evolves according to the geodesic formula $\mathbf{R}_t = \exp_{\mathbf{R}_0}\left(t\log_{\mathbf{R}_0}(\mathbf{R}_1)\right)$. For translation, the conditional probability path $p_t(\mathbf{x}_t \mid \mathbf{x}_0, \mathbf{x}_1)$ is defined by the linear interpolation $\mathbf{x}_t = (1-t)\mathbf{x}_0 + t\mathbf{x}_1$, which corresponds to the constant conditional vector field $u_t^{\mathbf{x}}(\mathbf{x}_t \mid \mathbf{x}_0, \mathbf{x}_1) = \mathbf{x}_1 - \mathbf{x}_0$ on $\mathbb{R}^3$.

The final objective is to regress the neural vector field $v_\theta = \left(v_\theta^{\mathbf{x}}, v_\theta^{\mathbf{R}}\right)$ to these target fields. Assuming an independent coupling between the prior $p_0$ and the data $p^*$, the loss is:

$$\mathcal{L}_{\mathrm{SE}(3)}(\theta) = \mathbb{E}_{\substack{\mathbf{T}_0 \sim p_0, \mathbf{T}_1 \sim p^* \\ t \sim \mathcal{U}(0,1)}}\Big[\|v_\theta^{\mathbf{x}}(\mathbf{T}_t, t) - u_t^{\mathbf{x}}(\mathbf{x}_t \mid \mathbf{x}_0, \mathbf{x}_1)\|^2$$
$$+ \|v_\theta^{\mathbf{R}}(\mathbf{T}_t, t) - u_t^{\mathbf{R}}(\mathbf{R}_t \mid \mathbf{R}_1)\|_{\mathrm{SO}(3)}^2\Big].$$

At inference, one samples an initial frame $\mathbf{T}_0 = (\mathbf{R}_0, \mathbf{x}_0)$ from the product prior $p_0 = \mathcal{U}_{\mathrm{SO}(3)} \times \mathcal{N}(\mathbf{0}, \mathbf{I})$ and numerically integrates the learned joint vector field $v_\theta$ to generate the final rigid frame configuration $\mathbf{T}_1 = (\mathbf{R}_1, \mathbf{x}_1)$.

### 2.2. Discrete Flows

While continuous approaches to flow matching on the categorical simplex exist (e.g., Eijkelboom et al., 2025; Davis et al., 2024), we follow Campbell et al. (2024) and handle generative modelling of discrete data with *continuous-time Markov chains* (CTMCs). Let $m$ denote a discrete random variable taking values in a finite vocabulary $\mathcal{V}_m$. Analogous to the continuous case, we aim to transform a sample $m_0$ from a tractable prior distribution $p_0$, e.g., a uniform distribution or a [MASK] state, to a data sample $m_1 \sim p^*$ via a probability path $p_t$ for $t \in [0, 1]$.

**CTMC Dynamics** The evolution of the marginal probability mass function $p_t$ over $\mathcal{V}_m$ is governed by the *Kolmogorov forward equation*. For a time-dependent transition *rate matrix* $\mathbf{Q}_t \in \mathbb{R}^{|\mathcal{V}_m| \times |\mathcal{V}_m|}$, the dynamics are given by:

$$\partial_t p_t(k) = \sum_{j \neq k} p_t(j)\mathbf{Q}_t(j, k) - p_t(k)\sum_{j \neq k}\mathbf{Q}_t(k, j),$$

where $\mathbf{Q}_t(j, k) \geq 0$ for $j \neq k$ represents the instantaneous rate of jumping from state $j$ to state $k$. This linear system serves as the discrete analogue to the continuity equation in continuous flow matching, with $\mathbf{Q}_t$ playing the role of the vector field $u_t$.

**Discrete Flow Matching** Constructing a generative model requires finding a rate matrix $\mathbf{Q}_t$ that generates a desired probability path $p_t$. Following the conditional flow matching paradigm, we define the marginal path as an expectation over conditional paths $p_t(m_t \mid m_1)$ anchored at the data sample $m_1$:

$$p_t(m_t) = \mathbb{E}_{m_1 \sim p^*} \left[ p_t(m_t \mid m_1) \right].$$

The conditional flow $p_t(\cdot | m_1)$ is typically chosen as a linear interpolation in probability space, such that $p_0(\cdot \mid m_1) = p_0(\cdot)$ and $p_1(\cdot \mid m_1) = \delta_{m_1}(\cdot)$, where $\delta$ is the Dirac delta. Campbell et al. (2024) demonstrate that the marginal rate matrix $\mathbf{Q}_t$ can be realized as the expectation of a *conditional rate matrix* $\mathbf{Q}_t(\cdot \mid m_1)$ which generates $p_t(\cdot \mid m_1)$:

$$\mathbf{Q}_t(j, k) = \mathbb{E}_{m_1 \sim p(m_1 \mid m_t = j)} \left[ \mathbf{Q}_t(j, k \mid m_1) \right].$$

Here, $\mathbf{Q}_t(j, k \mid m_1)$ is a closed-form rate matrix derived analytically to satisfy the conditional Kolmogorov equation for the chosen interpolant.

Unlike the continuous case, where we regress a vector field directly, learning the marginal rate matrix requires access to the posterior $p(m_1 \mid m_t = j)$. Consequently, we parameterise a denoising neural network $p_\theta(m_1 \mid m_t)$ to approximate the clean data distribution given a noisy state. The training objective is thus the cross-entropy loss:

$$\mathcal{L}_{\text{DFM}}(\theta) = \mathbb{E}_{\substack{m_1 \sim p^*, \, m_t \sim p_t(\cdot | m_1) \\ t \sim \mathcal{U}(0,1)}} \left[ -\log p_\theta(m_1 \mid m_t) \right].$$

At inference time, we construct the generative rate matrix $\mathbf{Q}_t^\theta$ using the learned denoiser: $\mathbf{Q}_t^\theta(j, k) = \mathbb{E}_{m_1 \sim p_\theta(\cdot | m_t = j)}[\mathbf{Q}_t(j, k \mid m_1)]$. New samples are generated by initialising $m_0 \sim p_0$ and simulating the CTMC trajectory defined by $\mathbf{Q}_t^\theta$.

## 3. Method

In this section, we start with a common representation of a molecule $\{(\mathbf{y}_j, h_j)\}_{j=1}^N$ as a point cloud of $N$ atoms, each with coordinates $\mathbf{y}_j \in \mathbb{R}^3$ and the atom type $h_j \in \mathcal{V}_a$. Here, $\mathcal{V}_a$ is the vocabulary of all atom types, including ions.

In Sections 3.1 and 3.2, we describe the *frame fragmentation*; its purpose is to reparametrise the molecule as $K$ rigid motifs $\{\mathcal{M}_i\}_{i=1}^K = \{(\mathbf{T}_i, m_i)\}_{i=1}^K$ where a frame $\mathbf{T}_i = (\mathbf{R}_i, \mathbf{x}_i) \in \text{SE}(3)$ has a rotation matrix $\mathbf{R}_i \in \text{SO}(3)$ and a translation vector $\mathbf{x}_i \in \mathbb{R}^3$ from the origin to the geometric centre of the motif. Each frame's rotation is defined relative to this motif's *exemplar fragment* $m_i \in \mathcal{V}_m$ from the *motif vocabulary* $\mathcal{V}_m$.

A fragment $m_i$ with $N_i$ atoms in this vocabulary describes the atom-level structure of the rigid motif, and is thus a tuple $m_i = (\mathbf{P}_i, \mathbf{h}_i, \mathcal{S}_i)$. Here, the fixed set of 3D coordinates of intra-fragment atoms $\mathbf{P}_i \in \mathbb{R}^{N_i \times 3}$ defines the motif's *canonical pose* centred at the origin. The corresponding types of intra-fragment atoms are denoted by $\mathbf{h}_i \in \mathcal{V}_a^{N_i}$. The third element, $\mathcal{S}_i \subset \text{SO}(3)$, represents the *discrete symmetry group of the motif*, i.e., the set of all rotations that, if applied to the pose $\mathbf{P}_i$, result in an indistinguishable molecular motif in 3D space.

We note that such frame-based representation is *invertible*: to recover the atom-level representation $\mathbf{Y}_i \in \mathbb{R}^{N_i \times 3}$ of atoms that corresponds to the rigid fragment $m_i$ with the frame $\mathbf{T}_i = (\mathbf{R}_i, \mathbf{x}_i)$ in a molecule in 3D space, one has to apply the rigid transformation to the motif's canonical pose:

$$\mathbf{Y}_i = \mathbf{P}_i \mathbf{R}_i + \mathbf{1}_{N_i} \mathbf{x}_i^\top \equiv \mathbf{P}_i \mathbf{S} \mathbf{R}_i + \mathbf{1}_{N_i} \mathbf{x}_i^\top, \; \forall \mathbf{S} \in \mathcal{S}_i$$

Under this rigid-frame parametrisation, generating *de novo* molecules with $K$ fragments can be seen as a task of sampling from the distribution on $\text{SE}(3)^K \times \mathcal{V}_m^K$. In Section 3.3, we formulate our generative framework MOTIFLOW.

### 3.1. Rigid-Motif Decomposition

To establish a motif vocabulary $\mathcal{V}_m$, we need to define a fragmentation scheme. Such a scheme has to satisfy several requirements, namely, (i) *rigidity*: fragments must be structurally rigid approximations (i.e., lacking internal rotatable bonds) to accurately represent all instances of a motif throughout the data, (ii) *non-degeneracy*: each fragment must possess at least three non-collinear points to define a frame $\mathbf{T} \in \text{SE}(3)$ and (iii) *tractability*, meaning that the frequency of each distinct class of $\mathcal{V}_m$ in the data has to be sufficiently high for learning the distribution over $\mathcal{V}_m^K$.

The fragmentation comprises two stages. We begin by identifying all sufficiently rigid structures: we preserve double and triple bonds, as well as all bonds within approximately *planar* rings and fused ring systems. Further, while we do cut the bonds between heavy atoms that are acyclic but adjacent to a cycle, unlike common fragmentation techniques (Jin et al., 2020; Maziarz et al., 2022), we do not cut bonds to hydrogens. This helps introduce fewer ill-defined rigid frames at the expense of having more unique motifs. After the chosen bonds are cut and the preliminary set of rigid motifs is established, we proceed by *pruning* them: analogous to common methods in molecular graph decomposition (e.g., Jin et al., 2020), we further fragment rigid motifs whose total number of occurrences across the dataset is less than $\alpha\%$ of the dataset size. We ablate different values of the hyperparameter $\alpha$ and its influence on the generation in Section 5.3. By default, we use $\alpha = 0.1$.

At this point, the only ill-defined frames are those that are collinear motifs (e.g., alkynes) or isolated atoms (e.g., Cl

atom cut from a $C_6H_5Cl$ chlorobenzene ring). To be able to uniquely define their rotation matrix $\mathbf{R} \in SO(3)$, similar to Prat et al. (2025), we start adding dummy atoms placed at a unit distance to the nearest non-collinear neighbours of this fragment in the original molecule until the orientation of the frame is locked and the rigid body is well-defined.

### 3.2. Canonicalisation and Vocabulary

Once the rigid motifs are defined, we proceed with constructing a vocabulary $\mathcal{V}_m$ where each element is a canonical descriptor of a chemical motif, formally defined as a tuple $m_i = (\mathbf{P}_i, \mathbf{h}_i, \mathcal{S}_i)$. For a given motif class $i$, the canonical fragment with its centred pose $\mathbf{P}_i \in \mathbb{R}^{N_i \times 3}$ and atom types $\mathbf{h}_i \in \mathcal{V}_a^{N_i}$ is chosen arbitrarily as the rigid motif's first occurrence during the dataset preprocessing and fixed. We compute $\mathcal{S}_i$ by identifying the set of all graph automorphisms $\Pi_i$, i.e., node permutations, that preserve chemical connectivity and element types in the motif. For each automorphism $\pi \in \Pi_i$, we derive the corresponding rotation matrix $\mathbf{R}_i^\pi$ that maps the exemplar onto its permuted self, i.e., $\mathbf{P}_i \approx \pi(\mathbf{P}_i)\mathbf{R}_i^\pi$. This explicitly encodes the rotational invariance of symmetric motifs, e.g., the indistinguishable orientations of cyclopropane $(CH_2)_3$, into the vocabulary.

We further assign a ground truth pose $\mathbf{T}_j = (\mathbf{R}_j, \mathbf{x}_j) \in SE(3)$ to each fragment instance $\mathcal{M}_j$ of type $m_i$ found in the data. The translation vector $\mathbf{x}_j$ is defined as the vector from the origin to the geometric centre of $\mathcal{M}_j$. We then compute a representative rotation $\mathbf{R}_j$ via the Kabsch algorithm (Kabsch, 1976) given any valid automorphism $\pi \in \Pi_i$ establishing the atom-wise correspondence:

$$\mathbf{R}_j = \underset{\mathbf{R} \in SO(3)}{\arg\min} \sum_{a=1}^{N_i} \left\| \mathbf{P}_{i,a}\mathbf{R} - \mathbf{Y}_{j,a}^\pi \right\|^2,$$

where $\mathbf{P}_{i,a}$ denotes the position of the $a$-th atom in $m_i$. This method allows for representing an arbitrary non-collinear molecule as a collection of well-defined rigid motifs from a tractable vocabulary $\mathcal{V}_m$. In Figure 2, we show for the QMUGS dataset that on average, such a fragment-based molecular parametrisation compresses the common all-atom and heavy-atom representations by factors of 3.4 and 1.8, respectively, without resorting to learning a latent space or requiring a computationally expensive reconstruction.

### 3.3. Multimodal Flow on $SE(3)^K \times \mathcal{V}_m^K$

Given the larger size of the motif vocabulary $\mathcal{V}_m$ compared to the common atom vocabulary $\mathcal{V}_a$, we opt not to approximate the discrete fragment classes in a one-hot continuous fashion, as commonly done in the literature (Hoogeboom et al., 2022; Cornet et al., 2024). Instead, we propose to natively handle the discrete and continuous supports of $m \in \mathcal{V}_m$ and $\mathbf{T} \in SE(3)$, respectively, in a *multimodal flow* (Campbell et al., 2024). Concretely, for a molecule with $K$ rigid motifs $\boldsymbol{\mathcal{M}} = \{\mathcal{M}^k\}_{k=1}^K$, MOTIFLOW models the

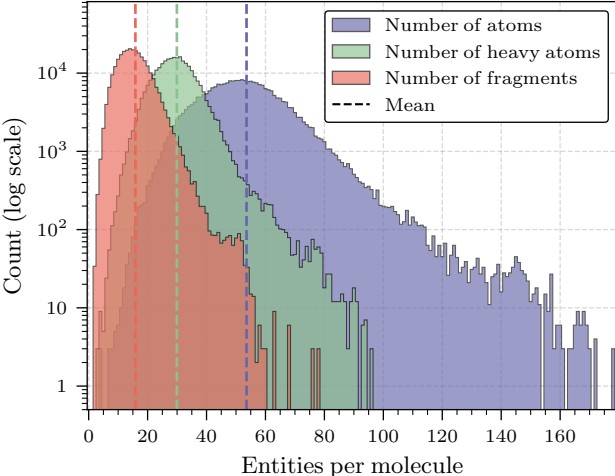

*Figure 2.* Comparison of molecular representations on QMUGS dataset (Isert et al., 2022). Fragmentation is reported for $\alpha = 0.1$.

following conditional flow[1], factorised over modalities and individual rigid-motif frames:

$$p_t(\boldsymbol{\mathcal{M}}_t \mid \boldsymbol{\mathcal{M}}_1) := \prod_{k=1}^K \underbrace{p_t(m_t^k \mid m_1^k)}_{\text{Discrete Flow}} \underbrace{p_t(\mathbf{T}_t^k \mid \mathbf{T}_1^k)}_{\text{SE(3) Flow}}.$$

This factorisation implies that the generative process for each rigid motif is independent *conditional* on the data sample $\boldsymbol{\mathcal{M}}_1$, allowing us to train the joint model by minimising a sum of modality-specific losses.

**Continuous Dynamics** For the geometric component, we independently apply the $SE(3)$ flow matching framework described in Section 2.1 to each of the $K$ rigid frames. The conditional probability path $p_t(\mathbf{T}_t^k \mid \mathbf{T}_1^k)$ is constructed via the product of the Euclidean interpolant for translations and the geodesic one for rotations. The training objective $\mathcal{L}_{SE(3)}$ is the sum over $K$ motifs of the regression loss between the network outputs and the target vector fields $u_t^{\mathbf{x}}$ and $u_t^{\mathbf{R}}$.

**Discrete Dynamics** For the motif types, we adopt the discrete flow from Section 2.2 using a *masking* prior. Specifically, we set the prior $p_0$ to be a Dirac delta on a special token, $m_0^k = [\text{MASK}]$ for all $k$. The conditional probability path interpolates linearly between this mask state and the true motif type $m_1^k$:

$$p_t(m_t^k \mid m_1^k) = (1-t)\delta_{[\text{MASK}]}(m_t^k) + t\delta_{m_1^k}(m_t^k).$$

To generate this path via a CTMC, we require the conditional rate matrix $\mathbf{Q}_t(j, l \mid m_1^k)$. For the masking interpolant, this matrix takes a simple analytic form where probability mass is transferred solely from $[\text{MASK}]$ to the target class $m_1^k$ (Campbell et al., 2024):

$$\mathbf{Q}_t(j, l \mid m_1^k) = \mathbb{I}(j = [\text{MASK}]) \cdot \mathbb{I}(l = m_1^k) \cdot \frac{1}{1-t}.$$

---

[1]For brevity, $p_t(\cdot \mid \cdot)$ refers to probability density and probability mass functions for continuous and discrete variables.

Intuitively, at any time $t \in (0, 1)$, a masked motif has a rate of $(1 - t)^{-1}$ to unmask to its true value $m_1^k$, while unmasked motifs remain fixed. To train the model, we employ a denoising network $p_\theta(m_1^k \mid \mathcal{M}_t)$ that predicts the categorical distribution over $\mathcal{V}_m$ given the noisy state of the entire molecule. The discrete objective $\mathcal{L}_{\text{DFM}}$ is the cross-entropy between the predicted logits and the true motif type $m_1^k$, summed over all masked motifs.

**Architecture** We parameterise the time-dependent vector fields and the discrete denoiser using a single unified neural network $v_\theta(\mathcal{M}_t, t)$. Our architecture builds upon the FOLDFLOW-BASE backbone (Bose et al., 2023), which utilises invariant point attention (IPA) (Jumper et al., 2021) to process 3D rigid frames. The network takes as input the noisy frames $\{\mathbf{T}_t^k\}_{k=1}^K$ and the embeddings of the partially masked motif tokens $\{m_t^k\}_{k=1}^K$. We introduce three modifications to adapt this architecture for multimodal molecular generation. First, we incorporate *self-conditioning* (Chen et al., 2023; Stärk et al., 2024) for the discrete modality: during training, with a probability $0.5$, we feed the model's own estimated clean motif types $\hat{m}_1$ back as input, improving coherence between the geometric and semantic features. Second, we augment the network's layers with triangular multiplicative updates (Jumper et al., 2021) on the pair of fragment representations to better capture the geometric constraints between rigid motifs, which, unlike residues in the protein backbones, are ordered arbitrarily. Finally, the network is equipped with an additional third prediction head that outputs logits over $\mathcal{V}_m$ for the discrete flow.

**Symmetries** This generative process has two kinds of physical symmetries. The first one, global $\text{SE}(3)$ equivariance, is guaranteed by the IPA backbone, ensuring that if the entire molecule is rotated and translated, the generated vector fields rotate and translate accordingly. Additionally, individual motifs $m_i$ may also possess non-trivial finite groups of rotational symmetries $\mathcal{S}_i$, and the true distribution is invariant to the choice of canonical pose representation from its orbit $\{\mathbf{P}_i \mathbf{S}_1, \ldots \mathbf{P}_i \mathbf{S}_{|\mathcal{S}_i|}\}$. Note that we set $\mathcal{S}_{[\text{MASK}]} \coloneqq \{\mathbf{I}\}$.

To handle this, we adopt a GEODIFF-style alignment strategy (Xu et al., 2022) for the loss computation. Instead of regressing towards a fixed canonical frame, we dynamically select the target rotation $\tilde{\mathbf{R}}_1^k$ from the symmetry orbit that is closest to the current noisy frame $\mathbf{R}_t^k$:

$$\tilde{\mathbf{R}}_1^k = \mathbf{S}^* \mathbf{R}_1^k, \quad \text{where} \quad \mathbf{S}^* = \underset{\mathbf{S} \in \mathcal{S}_k}{\arg\max} \operatorname{Tr}\left((\mathbf{R}_t^k)^\top \mathbf{S} \mathbf{R}_1^k\right).$$

Minimising the geodesic distance on $\text{SO}(3)$ corresponds to maximising the trace of the relative rotation matrix, which is computationally negligible as the finite symmetry groups of motifs are small. Empirically, we found that this alignment stabilises training at small times $t$, where the noisy state is close to the uninformed prior, preventing the flow from receiving conflicting gradients from equivalent but spatially distant symmetric targets.

We ablate the effects of the design choices and compare them with alternative options in Appendix A.4.4. Further implementation details are provided in Appendix A.2.

# 4. Related Work

**Fragment-Based Molecule Generation** In the task of molecular graph generation, several approaches to tokenising molecules into motifs exist, which can be classified into chemically inspired and data-driven ones. The former group (Jin et al., 2020; Maziarz et al., 2022; Jin et al., 2018; Lee et al., 2025) focuses on top-down separation of acyclic and cyclic parts in the molecule, while the latter group (Kong et al., 2022; Geng et al., 2023) adopts the bottom-up merging strategy starting from individual atoms. As an alternative to motif-based methods, Hetzel et al. (2025) proposed *scaffold-based* fragmentation, in which the molecular assembly starts from basic geometric shapes and attributes the chemical properties in the subsequent stages of generation. To the best of our knowledge, neither motif-based nor scaffold-based fragmentation has been widely explored in 3D. A notable exception is HIERDIFF (Qiang et al., 2023), which utilises a hierarchical diffusion framework to position coarse fragment nodes in 3D space. In contrast to our approach, which generates the full $\text{SE}(3)$ configuration of rigid motifs without auxiliary mechanisms, HIERDIFF relies on a learnable decoding procedure to resolve the specific identities and geometries of the fragments, thereby taking a fundamentally different route.

**3D Molecule Generation** Generation of spatial molecular structure is primarily tackled with flow-based (Satorras et al., 2021a; Igashov et al., 2024; Dunn & Koes, 2024) and autoregressive (Gebauer et al., 2022; Daigavane et al., 2024; Cheng et al., 2025) models on atom coordinates. A notable exception to these directions is the work of Pinheiro et al. (2024), where 3D molecules are represented as atomic densities on regular grids. Existing approaches to 3D molecular generation can also be roughly classified into two categories based on how they treat bonds. The seminal work of Hoogeboom et al. (2022) uses a lookup table to infer bond types from pairwise atom distances, which is the approach we adopt. Recently, a line of work (Reidenbach et al., 2025; Vignac et al., 2023; Irwin et al., 2024; Peng et al., 2023) has proposed a joint modelling of the 2D molecular graph topology and 3D atom coordinates. While they show improvement over point cloud approaches, we perform our experiments without modelling the graph structure beyond intra-motif connectivity, i.e., beyond the bond structure within the fragments; therefore, we do not directly compare our method to these approaches.

**Rigid-body Generation** Parametrising protein residues as rigid frames has been originally introduced in the seminal

work of Jumper et al. (2021) and received a widespread adoption in the subsequent methods for protein structure prediction and design (Watson et al., 2023; Yim et al., 2023; Bose et al., 2023). Generative modelling with rigid frames outside the protein application, however, is limited. A concurrent work (Prat et al., 2025) explores rigid-fragment SE(3) diffusion for molecular docking, which is an orthogonal task to ours. *De novo* generation of 3D structure of general molecules from rigid fragments thus remains underexplored, which is a gap the present paper aims to fill.

## 5. Experiments

**Tasks and Datasets**  In Section 5.1, we consider the task of unconditional generation on two common benchmarks, QM9 (Ramakrishnan et al., 2014) and GEOM-DRUGS (Axelrod & Gomez-Bombarelli, 2022). The former dataset contains 134k small organic molecules with up to 29 atoms in total, with a maximum of 9 heavy atoms. The latter, larger-scale dataset of molecular conformers comprises 430k medium-sized molecules, with up to 181 atoms and an average of 44.4 atoms per molecule. For both datasets, we follow the same setup as in previous works (Hoogeboom et al., 2022; Cornet et al., 2024; Xu et al., 2022). We then proceed to the two conditional experiments on QM9 in Section 5.2, demonstrating our method's ability to generate molecules with desired properties. We conclude by studying the proposed rigid-motif fragmentation strategy and its modifications on the two datasets containing larger molecules, which is the principal use case of our method: the aforementioned GEOM-DRUGS and QMUGS (Isert et al., 2022). The QMUGS dataset contains 665k large drug-like molecules, with up to 100 heavy atoms; we use its subset of 300k conformations for our experiments. For experimental details and extended results, see Appendix A.3 and A.4.

**Baselines**  We compare the performance of MOTIFLOW to models within the same paradigm of inferring bonds from interatomic distances (Hoogeboom et al., 2022; Wu et al., 2022; Xu et al., 2023; Song et al., 2023; Cornet et al., 2024; Song et al., 2024). Further details are in Appendix A.3.1.

### 5.1. Unconditional Generation

For this task, we follow Cornet et al. (2024) and sample $10^4$ molecules across 3 seeds, reporting the mean and standard deviation. For each baseline with fixed-step solvers, we provide the step configuration that yielded the best molecular stability for QM9 and atom stability for GEOM-DRUGS, among those reported in the respective papers.

**Metrics**  Following the established practice (Hoogeboom et al., 2022; Song et al., 2023), we report the percentages of stable atoms A and molecules M, as well as valid V and valid and unique V×U molecules computed on RDKit (Landrum et al., 2025). Since for larger GEOM-DRUGS, the connectivity of the generated molecules is commonly re-

*Table 1.* Results of unconditional generation on QM9.

| Method | Steps | Stability (↑) | | Val. / Uniq. (↑) | |
|---|---|---|---|---|---|
| | | A, % | M, % | V, % | V × U, % |
| Data | – | 99.0 | 95.2 | 97.7 | 97.7 |
| EDM
Hoogeboom et al. (2022) | $10^3$ | 98.7 | 82.0 | 91.9 | 90.7 |
| EDM-BRIDGE
Wu et al. (2022) | $10^3$ | 98.8 | 84.6 | 92.0 | 90.7 |
| GEOLDM
Xu et al. (2023) | $10^3$ | $98.9_{\pm.1}$ | $89.4_{\pm.5}$ | $93.8_{\pm.4}$ | $92.7_{\pm.5}$ |
| EQUIFM
Song et al. (2023) | – | $98.9_{\pm.1}$ | $88.3_{\pm.3}$ | $94.7_{\pm.4}$ | $\mathbf{93.5}_{\pm.3}$ |
| END
Cornet et al. (2024) | $10^3$ | $98.9_{\pm.0}$ | $89.1_{\pm.1}$ | $94.8_{\pm.1}$ | $92.6_{\pm.2}$ |
| EDM*
Cornet et al. (2024) | $10^3$ | $98.4_{\pm.0}$ | $85.3_{\pm.3}$ | $93.5_{\pm.1}$ | $91.9_{\pm.1}$ |
| GEOBFN
Song et al. (2024) | $10^3$ | $99.1_{\pm.1}$ | $90.9_{\pm.2}$ | $95.3_{\pm.1}$ | $93.0_{\pm.1}$ |
| MOTIFLOW | $10^2$ | $99.1_{\pm.1}$ | $\mathbf{92.6}_{\pm.5}$ | $95.3_{\pm.6}$ | $86.3_{\pm.9}$ |

ported to be more challenging, while practically all samples are unique, we follow Cornet et al. (2024) and report the percentage of valid and connected V×C molecules instead. In line with the baselines, we also do not report molecular stability for GEOM-DRUGS, as it was found to be non-informative when bonds are inferred from interatomic distances (Hoogeboom et al., 2022; Song et al., 2023).

**QM9**  Main results are provided in Table 1. The QM9 benchmark is fairly saturated on unconditional generation and thus mainly reported for completeness. Overall, MOTI-FLOW performs on par with methods that use $10\times$ the number of generation steps: we obtain higher molecular stability and lower uniqueness than the best-performing EQUIFM (Song et al., 2023), both being the consequence of using rigid motifs as larger building blocks on small molecules; this trade-off is controllable via, e.g., the sampling temperature in the discrete flow.

**GEOM-DRUGS**  Main results[2] are presented in Table 2. On this dataset of larger, more realistic, and challenging molecules, MOTIFLOW significantly outperforms the baselines. We note that our use of rigid motifs as larger building blocks also allows us to ameliorate errors in atom valencies that the true atom-level data exhibits when assessed by the bond lookup mechanism of Hoogeboom et al. (2022).

### 5.2. Conditional Generation

With the conditional experiments, we seek to answer two primary questions that arise naturally from the proposed change in molecular representations:

(i) *Fine-grained conditioning*: does the motif-based representation perform competitively at generation conditioned

---

[2]Cornet et al. (2024) have conflicting V×C results on GEOM-DRUGS for their END model reported in the appendix and main body of their paper; we resort to results provided in the main body.

*Table 2.* Results of unconditional generation on GEOM-DRUGS. [‡]Results are obtained by Cornet et al. (2024).

| Method | Steps | Stable A, % ($\uparrow$) | V $\times$ C, % ($\uparrow$) |
|---|---|---|---|
| Data | – | 86.5 | 99.0 |
| GEOLDM Xu et al. (2023) | 1000 | 84.4 | 45.8[‡] |
| EQUIFM Song et al. (2023) | – | 84.1 | – |
| END Cornet et al. (2024) | 100 | $87.2_{\pm.1}$ | $73.7_{\pm.4}$ |
| EDM* Cornet et al. (2024) | 250 | $85.4_{\pm.0}$ | $61.4_{\pm.6}$ |
| GEOBFN Song et al. (2024) | 1000 | 85.6 | – |
| MOTIFLOW | 100 | $\mathbf{95.0_{\pm.0}}$ | $\mathbf{81.2_{\pm.3}}$ |

on the *atom-level* information, despite only modelling it implicitly?

(ii) *Coarse-grained conditioning*: does the use of fragments lead to better generation of desired *substructures*?

We adopt the tasks for both scenarios from the previous works (Cornet et al., 2024; Bao et al., 2023). For the fine-grained level, we condition on the *atom composition* $\mathbf{c} = (c_1, \ldots, c_{|\mathcal{V}_a|}) \in \mathbb{R}^{|\mathcal{V}_a|}$, where $c_j$ is the number of atoms of the type $j$ that the desired sampled molecule is required to have. Following Cornet et al. (2024), we generate 10 samples for each unique target atom composition from the validation and test sets across 3 seeds, and report the percentage of matched compositions. For the coarse-grained level, we condition on the *substructural features*: concretely, we use a molecular fingerprint $\mathbf{c} = (c_1, \ldots, c_F) \in \{0, 1\}^F$, each entry of which indicates the presence or absence of a certain substructure in the molecule. We use OpenBabel (O'Boyle et al., 2011) to generate the fingerprints for the test set, and evaluate the generation by computing Tanimoto similarity between the fingerprints of generated molecules and those from the test set, which are injected as conditions to the model. To ensure a fair comparison with the baselines in both tasks, we follow the strategy of Cornet et al. (2024): while many techniques for guiding flow models exist (Ho & Salimans, 2022; Schiff et al., 2025), in this section, we directly use the conditional model $v_\theta(\mathcal{M}_t, t, \mathbf{c})$ by adding the conditioning information $\mathbf{c}$ directly to the input. We denote this version as CMOTIFLOW.

The results are summarised in Table 3, with extended results and qualitative examples in Appendix A.4. The tasks are concisely summarised in Figure 3. CMOTIFLOW outperforms the baselines, achieving better adherence to conditioning information at both levels of molecular granularity while requiring fewer generation steps.

**5.3. Ablations**

The purpose of this section is to further study the effects of the rigid-motif decomposition on 3D molecular generation. As noted by existing literature (e.g., Hetzel et al., 2025), the motif-based molecular graph decomposition offers a trade-off: while many larger functional groups are present in the data, adding larger motifs to the vocabulary leads, on

*Table 3.* Results of the two conditional generation tasks on QM9.

| Method | Steps | COMPOSITION Matching, % ($\uparrow$) | SUBSTRUCTURE Tanimoto Sim. ($\uparrow$) |
|---|---|---|---|
| CEDM Bao et al. (2023) | 1000 | – | $.671_{\pm.004}$ |
| EEGSDE Bao et al. (2023) | 1000 | – | $.750_{\pm.003}$ |
| CEDM* Cornet et al. (2024) | 500 | $76.2_{\pm0.6}$ | $.669_{\pm.001}$ |
|  | 1000 | $75.5_{\pm0.5}$ | $.673_{\pm.002}$ |
| CEND Cornet et al. (2024) | 500 | $91.5_{\pm0.8}$ | $.825_{\pm.001}$ |
|  | 1000 | $91.0_{\pm0.9}$ | $.828_{\pm.001}$ |
| CMOTIFLOW | 100 | $\mathbf{95.4_{\pm0.5}}$ | $\mathbf{.862_{\pm.002}}$ |

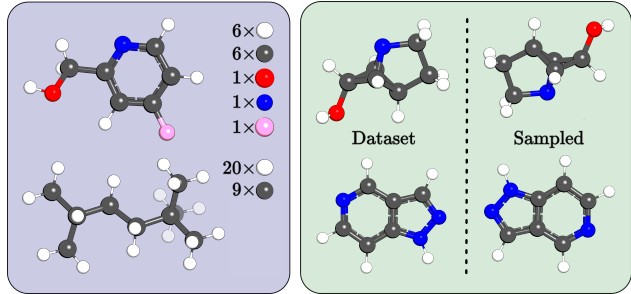

*Figure 3.* Examples of results for the conditional tasks on QM9: atom composition (*left*) and fingerprint substructure (*right*).

average, to less frequent classes observed during training, which could make generalisation to uncommon substructures challenging. At the same time, we hypothesise that in 3D, larger rigid motifs, if sufficiently frequent, bring about the benefits of the fragment-based generation, which were empirically established in Sections 5.1 and 5.2.

To verify this, we consider several variations of the rigid-motif decomposition. In the NO RINGS variant, we only preserve bonds to hydrogen atoms, as well as double and triple bonds. This configuration allows us to evaluate the performance of the finer-grained vocabulary with fewer rare fragments. On the opposite side of the spectrum, we consider our main decomposition introduced in Section 3.1, which preserves approximately planar rings and fused ring systems. For this PLANAR RINGS fragmentation, we consider three thresholds (including our base one, $\alpha = 0.1$) for the minimal frequency of a fragment relative to the dataset size; i.e., a smaller threshold corresponds to a larger vocabulary with more rare motifs, while with a larger threshold, they are decomposed further into finer components. An example of a molecule from GEOM-DRUGS under different rigid-motif decompositions is provided in Figure 4. Detailed information on the setup can be found in Appendix A.4.4.

First, we ablate the performance of each fragmentation strategy at the unconditional generation on GEOM-DRUGS and QMUGS; as it is for the main unconditional experiments, we evaluate the metrics based on $10^4$ samples obtained with

3 seeds; results are in Table 4. Our hypothesis is largely confirmed: all rigid-motif vocabularies that treat planar rings as distinct classes yield superior atom-stability performance compared to finer fragmentation, with the two smaller threshold configurations scoring best.

To assess the extent to which uncommon motifs are purposefully sampled during generation, we analyse the motif set of generated molecules. We first formally define *common* and *uncommon* motifs as those that exceed the base occurrence threshold $\alpha = 0.1$ and those that do not reach its frequency in the data, respectively. For each group, we then compute the total counts of its constituent motifs, normalised to the total number of molecules, independently for the training and generated sets. The ratio of these normalised counts, if the model accurately represents the underlying distribution of motifs, should be close to 1 for both common and uncommon motifs. Figure 5 shows that, while common motifs are consistently generated with close to the true occurrence, the uncommon motifs are notably oversampled for the finer NO RINGS strategy. Further, we observe an escalating undersampling behaviour for PLANAR RINGS strategy with the decrease in threshold $\alpha$, confirming the expected trade-off between the vocabulary size and uncommon motif coverage (Hetzel et al., 2025). We thus conclude that frequency-based fragmentation with planar ring systems at a relatively high $\alpha = 0.1$ offers the optimal trade-off, significantly improving generation over conventional atom-based baselines and other rigid decomposition strategies.

*Table 4.* Ablation results for different fragmentation strategies. Metrics are reported for sampling with 100 reverse steps.

| Strategy | QMUGS | | | GEOM-DRUGS | | |
|---|---|---|---|---|---|---|
| | $|\mathcal{V}_m|$ | Stability A, % (↑) | V × C % (↑) | $|\mathcal{V}_m|$ | Stability A, % (↑) | V × C % (↑) |
| NO RINGS | 49 | $85.5_{\pm1.8}$ | $\mathbf{85.6}_{\pm.3}$ | 39 | $86.7_{\pm1.8}$ | $\mathbf{82.9}_{\pm.4}$ |
| PLANAR RINGS 0.5% | 96 | $95.5_{\pm0.1}$ | $84.0_{\pm.5}$ | 81 | $95.2_{\pm0.1}$ | $82.1_{\pm.3}$ |
| PLANAR RINGS 0.1% | 253 | $\mathbf{96.1}_{\pm0.0}$ | $83.1_{\pm.4}$ | 202 | $95.0_{\pm0.0}$ | $81.2_{\pm.3}$ |
| PLANAR RINGS 0.01% | 1184 | $95.3_{\pm0.0}$ | $80.4_{\pm.5}$ | 867 | $\mathbf{96.3}_{\pm0.0}$ | $82.4_{\pm.3}$ |

## 6. Conclusion

In this work, we introduced MOTIFLOW, a novel generative framework for 3D molecules that operates on *rigid motifs* rather than individual atoms. By combining the rigid-motif decomposition strategy with the multimodal flow matching objective, we generate drug-like molecules via flows on the SE(3) manifold coupled with discrete categorical flows. Our empirical evaluation on standard benchmarks demonstrates that this higher-level representation yields improved stability on larger molecules compared to established all-atom methods, while offering more concise molecular

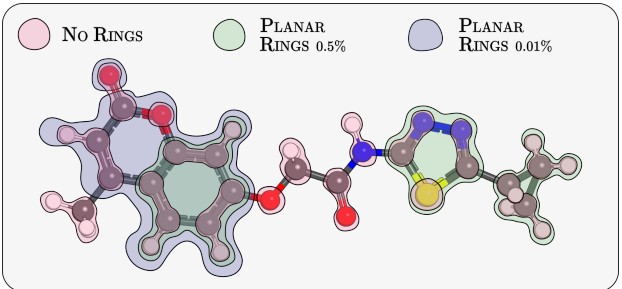

*Figure 4.* Resulting sets of rigid motifs for a GEOM-DRUGS molecule $C_{17}H_{15}N_3O_4S$ under different fragmentation strategies.

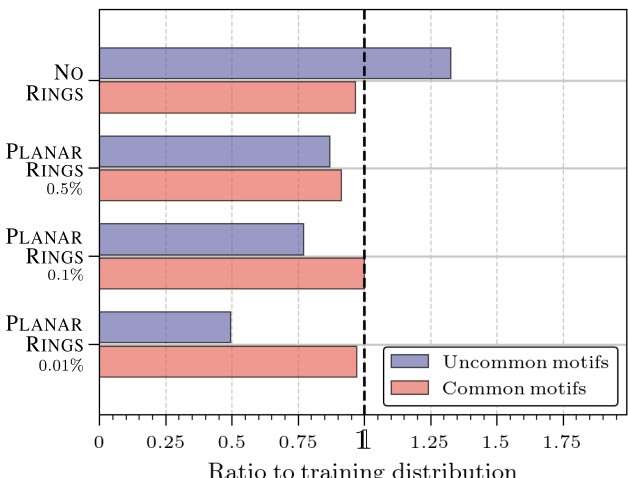

*Figure 5.* Comparison of sampled motif types to their frequencies in the training distribution of QMUGS. A ratio of 1 is optimal.

representations as well as significant advantages in sampling efficiency and conditional generation capabilities.

**Limitations and Future Work** This method, while extending *motif-based* molecular graph generation to the 3D space, also inherits its limitations. Most notably, the reliance on a predefined vocabulary introduces a trade-off between vocabulary size and generalisation: as the vocabulary grows to capture larger motifs, the frequency of individual classes drops, leading to lower coverage of uncommon substructures during sampling. An alternative, *scaffold-based* generation (Hetzel et al., 2025), however, is non-trivial in 3D due to the vastly different spatial geometries of fragments of the same shape, and thus its integration into our method constitutes an exciting direction for future work. Of potential interest could also be an extension of our method to models that jointly generate 3D and 2D molecular structures (e.g., Reidenbach et al., 2025), thereby bridging motif-based generation with explicit modelling of all bonds.

**Broader Impact** Generative models for molecules have the potential to accelerate *in-silico* discovery and the design of novel drugs and materials, but they also carry potential dangers, as such models could be misused for designing chemicals with socially adverse properties.

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

# A. Supplementary Material

## A.1. Flow Matching in Euclidean Spaces

The task of generative modelling can be seen as transporting a sample $\mathbf{x}_0 \sim p(\mathbf{x})$ in $\mathbb{R}^d$ from a tractable prior distribution $p$ to a data sample $\mathbf{x}_1 \sim p^*(\mathbf{x})$ of the unknown data distribution $p^*$. In this formulation, the goal is thus to construct a *probability density path* $p_t$, $t \in [0,1]$, such that $p_0 \approx p$, $p_1 \approx p^*$. *Flow matching* (Lipman et al., 2022; Albergo & Vanden-Eijnden, 2022; Liu et al., 2022) approaches it by introducing a *flow*, i.e., an *interpolant*, $\phi_t(\mathbf{x}_0)$, such that $\mathbf{x}_t = \phi_t(\mathbf{x}_0) \sim p_t$ if $\mathbf{x}_0 \sim p_0$. One can model $\phi_t(\mathbf{x}_0)$ as a solution of an ordinary differential equation (ODE) with a time-varying *vector field* $u_t(\mathbf{x}_t): \mathbb{R}^d \times [0,1] \to \mathbb{R}^d$ and the initial condition $\phi_0(\mathbf{x}_0) = \mathbf{x}_0$:

$$\frac{d\mathbf{x}_t}{dt} = u_t(\mathbf{x}_t) \quad \text{and} \quad \phi_1(\mathbf{x}_0) = \mathbf{x}_0 + \int_0^1 u_t(\mathbf{x}_t)\, dt.$$

The training objective arises from regressing a neural network $v_\theta: \mathbb{R}^d \times [0,1] \to \mathbb{R}^d$ to the vector field $u_t$:

$$\mathcal{L}_{\text{FM}}(\theta) = \mathbb{E}_{\substack{\mathbf{x}_t \sim p_t(\mathbf{x}) \\ t \sim \mathcal{U}(0,1)}} \|v_\theta(\mathbf{x}_t, t) - u_t(\mathbf{x}_t)\|^2.$$

Computing this loss requires access to intractable $u_t(\mathbf{x})$ and $p_t$. It can be shown (e.g., Lipman et al., 2022) that optimising $\mathcal{L}_{\text{FM}}$ is equivalent in expectation to optimising the *conditional flow matching* (CFM) objective $\mathcal{L}_{\text{CFM}}$:

$$\mathcal{L}_{\text{CFM}}(\theta) = \mathbb{E}_{\substack{\mathbf{z} \sim p(\mathbf{z}),\, \mathbf{x}_t \sim p_t(\mathbf{x}|\mathbf{z}) \\ t \sim \mathcal{U}(0,1)}} \|v_\theta(\mathbf{x}_t, t) - u_t(\mathbf{x}_t \mid \mathbf{z})\|^2,$$

where $u_t(\mathbf{x} \mid \mathbf{z}): \mathbb{R}^d \times [0,1] \to \mathbb{R}^d$ is a conditional vector field that generates a conditional probability path $p_t(\mathbf{x} \mid \mathbf{z})$. That is, instead of regressing over the marginal vector field, we regress over conditional vector fields. The conditioning is commonly done on the data point $\mathbf{z} = \mathbf{x}_1$ or on a source-target coupling $\mathbf{z} = (\mathbf{x}_0, \mathbf{x}_1)$ (Tong et al., 2024).

## A.2. Implementation Details

**Geometric Matching and Canonicalisation**   To ensure geometric robustness, we quantify the dimensionality of motifs using singular value decomposition (SVD) on the centred coordinate matrix $\mathbf{P} \in \mathbb{R}^{N \times 3}$. Let $\sigma_1 \geq \sigma_2 \geq \sigma_3$ denote the singular values of $\mathbf{P}$. We classify a motif as *linear* if $\sigma_2 < 0.05 \cdot \sigma_1$, and *planar* if $\sigma_3 < 0.05 \cdot \sigma_1$. This check determines whether dummy atoms are required to lock the coordinate frame. During vocabulary construction, a candidate fragment instance matches an existing vocabulary class if its structural similarity falls within a specified tolerance. We align the instance to the exemplar using the Kabsch algorithm with atom-wise weights $w$, setting $w = 1.0$ for heavy atoms and $w = 0.1$ for hydrogens and dummy atoms. This weighting ensures the alignment is driven by the rigid heavy-atom scaffold, while still utilising lighter atoms to uniquely define the orientation of frames with collinear or planar heavy-atom cores. A match is accepted if the weighted root mean square deviation (RMSD) is $\leq 1.0$ Å and the maximum single-atom deviation is $\leq 1.0$ Å. Since we pre-select naturally rigid chemical moieties, these rather loose thresholds are not sensitive to minor structural noise (e.g., bond length vibrations); rather, they ensure that, e.g., distinct *geometric stereoisomers*, such as the *(E)*- and *(Z)*-isomers of alkenes that share the same graph topology but differ in rigid 3D arrangement, are correctly assigned to separate vocabulary entries.

**Architecture**   Our model architecture is based on the FOLDFLOW-BASE backbone (Bose et al., 2023), which utilises Invariant Point Attention (IPA) (Jumper et al., 2021) to process 3D rigid frames. To adapt this protein-specific architecture for general rigid-motif molecular generation, we make several key modifications. We reduce the hidden dimension per attention head to 64 for each of 8 heads, resulting in a compact model size of 12.4 million parameters for the unconditional model, compared to 17 million in the original FOLDFLOW-BASE. The conditional variant, CMOTIFLOW, utilises 13.7 million parameters due to the additional projection layers required for processing conditioning signals.

**Embeddings and Input Processing**   The network accepts a set of noisy rigid frames $\mathbf{T}_t$ and discrete motif tokens $m_t$. The node embeddings are initialized by projecting the motif token embeddings, sinusoidal timestep embeddings, and, during training, with 50% probability, the self-conditioning embeddings derived from the previous step's prediction.

Unlike protein models that rely on residue index offsets for edge initialization, our motifs have no intrinsic linear ordering. Consequently, we replace sequence-based edge biases with purely geometric information. We compute pairwise Euclidean

distances between the centers of rigid motifs and encode them using Gaussian Radial Basis Functions (RBFs) with 64 basis functions. These geometric features are concatenated with the cross-concatenated node features to form the initial edge embeddings. The point cloud of fragments is treated as a fully connected graph, i.e., each pair of fragments is assigned an edge embedding.

**Backbone and Updates**   The backbone consists of 4 blocks, each containing an IPA layer, transition modules, and 2 transformer encoder layers. To better capture the global geometric constraints of an unordered molecule, we augment the standard IPA blocks with triangle multiplicative updates (Jumper et al., 2021). Specifically, we apply both outgoing and incoming triangle updates to the edge representations before the attention mechanism in every block.

For the geometry updates, we employ a split-head design: the rotational update is predicted via a linear layer initialized to zero, while the translational update is predicted via a small 2-layer MLP.

**Conditioning**   For conditional generation tasks, we follow the conditioning strategy of Cornet et al. (2024). The conditioning signal $c$, i.e., either atom composition or substructure fingerprint, is encoded into a global context vector via a dedicated MLP. For composition conditioning, we learn embeddings for each atom type and compute a weighted sum based on the target counts before passing it to the MLP. This global context is injected into the network at three points: (i) concatenated to the initial node embeddings, (ii) added as a bias to the node representations within each IPA block, and (iii) concatenated to the final node representations before the discrete readout head.

**Training**   We train the model using the flow matching objective described in Section 3. For the rotational component of the SE(3) flow, we use the exponential rate scheduler (Bose et al., 2023) with the factor of 10, following Yim et al. (2023). In each experiment, we use learning rate of $10^{-4}$.

### A.3. Experimental Details

#### A.3.1. BASELINE DETAILS

In this section, we briefly overview the baseline methods used for comparison in our experiments. All baselines are atom-based generative models that generate 3D molecules by determining the type and coordinates of individual atoms.

**EDM (Hoogeboom et al., 2022)** is a seminal work that introduces a score-based generative model operating directly on the continuous coordinates and discrete atom types of the molecule. It utilizes an E(3)-equivariant graph neural network (Satorras et al., 2021b) to learn a denoising process that reverses a diffusion process, which transforms data into Gaussian noise. The model ensures that the generated likelihood is invariant to rotations and translations.

**EDM-BRIDGE (Wu et al., 2022)** improves upon the EDM framework by replacing the standard Gaussian prior with an informative prior that contains structural or physical information. It learns a bridge process that connects this informative prior to the data distribution, utilizing Lyapunov functions to guide the generation and improve stability and validity.

**GEOLDM (Xu et al., 2023)** is a latent diffusion framework for 3D molecules. Unlike EDM, which operates in the data space, GEOLDM first compresses molecular structures into a low-dimensional, continuous latent space using an E(3)-invariant autoencoder. A diffusion model is then trained in this latent space, enabling more efficient sampling and the capture of high-level geometric features.

**EQUIFM (Song et al., 2023)** applies the flow matching framework to 3D molecular generation. It proposes a hybrid probability transport path: it uses optimal transport for continuous atomic coordinates to generate straight trajectories, and a conditional probability path for discrete atom types. This formulation allows significantly faster sampling speeds than diffusion-based equivalents while maintaining SE(3) equivariance.

**END (Cornet et al., 2024)** is a recent diffusion-based method that introduces a learnable forward process. Unlike standard diffusion models that use a fixed, predefined corruption process (e.g., adding Gaussian noise), END parameterises the forward process with a time- and data-dependent transformation that is equivariant to rigid transformations. This flexibility allows the model to learn a more optimal degradation and restoration path for molecular geometries.

**GEOBFN (Song et al., 2024)** adapts the Bayesian flow network (Graves et al., 2023) framework to 3D geometry. Instead of adding noise to the data directly, GEOBFN operates on the parameters of the data distribution (e.g., mean and variance for coordinates, probabilities for atom types). It updates these parameters iteratively using Bayesian inference and a neural

network, unifying the generation of discrete and continuous modalities in a differentiable parameter space.

### A.3.2. EVALUATION METRICS

We adopt the evaluation framework and metric definitions from previous works (Lee et al., 2025; Cornet et al., 2024; Qiang et al., 2023). All metrics are implemented using `rdkit` (Landrum et al., 2025). Consistent with the reference methodology (Lee et al., 2025; Cornet et al., 2024), unless otherwise noted, raw generated coordinate samples are converted into molecular objects using `OpenBabel` (O'Boyle et al., 2011) and property metrics are calculated exclusively on samples that are both valid and connected.

- **Atom and Molecule Stability**: atom stability is determined by valency constraints; an atom is stable if its charge is 0. Molecular stability requires every atom in the structure to be stable. Bond types are inferred from pairwise atomic distances using the lookup table method from Hoogeboom et al. (2022).

- **Validity and Connectivity**: *validity* measures the percentage of generated samples that successfully pass `rdkit`'s parsing and sanitisation checks. However, standard sanitisation does not penalise disconnected molecules. To address this, particularly for larger molecules in GEOM-DRUGS and QMUGS datasets, we report *connectivity*, which verifies that a valid molecule consists of a single connected component.

- **Uniqueness**: this metric reports the percentage of valid samples that possess a unique SMILES string (Weininger, 1988). Following standard practice for larger molecules in GEOM-DRUGS and QMUGS datasets, we omit uniqueness from those results as generated samples are almost invariably unique.

- **Total Variation (TV)**: to measure the distributional fidelity of atom and bond types, we compute the total variation. This is defined as the mean absolute error between the marginal distributions of atom and bond types in the training set and those in the generated samples.

- **Strain Energy**: this metric assesses the geometric quality of the molecules. It is calculated as the difference between the energy of the generated structure and the energy of its relaxed conformation. The relaxation is performed using the MMFF force field within `rdkit` (Landrum et al., 2025).

- **Chemical Properties (SA, QED)**: we report a couple of standard molecular descriptors:
    - **SA**: the Synthetic Accessibility score (Ertl & Schuffenhauer, 2009) estimates the difficulty of synthesising a molecule. We follow Cornet et al. (2024) and normalise this score to the interval $[0, 1]$, where 1 indicates high synthesisability (easy to synthesise) and 0 indicates low synthesisability.
    - **QED**: the Quantitative Estimation of Drug-likeness.

### A.4. Extended Experimental Results

#### A.4.1. UNCONDITIONAL GENERATION

**QM9**   We provide the extended results in Table 5.

We observed no improvement in generation when setting more than 100 steps in the sampling process, consistent with the evidence that flow models generally require fewer steps than diffusion models (Lipman et al., 2022). We used a linear schedule for sampling temperature in the discrete flow, with a minimum of 1.0 and a maximum of 1.5. For all QM9 experiments, we set the remasking probability $\eta$ (Campbell et al., 2024) in the discrete flow to zero, thus disabling it.

On QM9, we train all models for 1000 epochs.

**GEOM-DRUGS**   We follow Hoogeboom et al. (2022) and opt not to report the molecular stability, neither inferring it via their lookup tables nor via `OpenBabel`, since the latter introduces significant bias and errors (Reidenbach et al., 2025). We used a fixed sampling temperature of 0.1 in the discrete flow. For GEOM-DRUGS, we set the remasking probability in the discrete flow to $\eta = 1.5$.

We train the model for 135 epochs. The extended results are provided in Table 6.

*Table 5.* Extended results at unconditional generation on QM9 with additional metrics from Qiang et al. (2023). [†]Song et al. (2023) uses Dopri5 adaptive solver.

| Method | Steps | Stability (↑) | | Val. / Uniq. (↑) | | TV (↓) | | Str. En. (↓) | SA (↑) | QED (↑) |
|---|---|---|---|---|---|---|---|---|---|---|
| | | A, % | M, % | V, % | V × U, % | $A\left[10^{-2}\right]$ | $B\left[10^{-3}\right]$ | $\Delta E\left[\frac{kcal}{mol}\right]$ | | |
| Data | – | 99.0 | 95.2 | 97.7 | 97.7 | – | – | 7.7 | 0.63 | 0.46 |
| EDM Hoogeboom et al. (2022) | 1000 | 98.7 | 82.0 | 91.9 | 90.7 | – | – | – | – | – |
| EDM-BRIDGE Wu et al. (2022) | 1000 | 98.8 | 84.6 | 92.0 | 90.7 | – | – | – | – | – |
| GEOLDM Xu et al. (2023) | 1000 | $98.9_{\pm.1}$ | $89.4_{\pm.5}$ | $93.8_{\pm.4}$ | $92.7_{\pm.5}$ | 1.6 | 1.3 | 10.4 | – | – |
| EQUIFM Song et al. (2023) | $-^{\dagger}$ | $98.9_{\pm.1}$ | $88.3_{\pm.3}$ | $94.7_{\pm.4}$ | $93.5_{\pm.3}$ | – | – | – | – | – |
| END Cornet et al. (2024) | 50 | $98.6_{\pm.0}$ | $84.6_{\pm.1}$ | $92.7_{\pm.1}$ | $91.4_{\pm.1}$ | $1.5_{\pm.1}$ | $1.9_{\pm.4}$ | $12.1_{\pm.3}$ | $0.60_{\pm.00}$ | $0.46_{\pm.00}$ |
| | 100 | $98.8_{\pm.0}$ | $87.4_{\pm.2}$ | $94.1_{\pm.0}$ | $92.3_{\pm.2}$ | $1.3_{\pm.0}$ | $1.8_{\pm.3}$ | $10.6_{\pm.2}$ | $0.61_{\pm.00}$ | $0.46_{\pm.00}$ |
| | 250 | $98.9_{\pm.1}$ | $88.8_{\pm.5}$ | $94.7_{\pm.2}$ | $92.6_{\pm.1}$ | $1.2_{\pm.1}$ | $0.8_{\pm.2}$ | $9.6_{\pm.2}$ | $0.62_{\pm.00}$ | $0.46_{\pm.00}$ |
| | 500 | $98.9_{\pm.0}$ | $88.8_{\pm.4}$ | $94.8_{\pm.2}$ | $92.8_{\pm.2}$ | $1.2_{\pm.1}$ | $0.8_{\pm.5}$ | $9.5_{\pm.1}$ | $0.62_{\pm.00}$ | $0.46_{\pm.00}$ |
| | 1000 | $98.9_{\pm.0}$ | $89.1_{\pm.1}$ | $94.8_{\pm.1}$ | $92.6_{\pm.2}$ | $1.2_{\pm.1}$ | $0.8_{\pm.5}$ | $9.3_{\pm.1}$ | $0.62_{\pm.00}$ | $0.46_{\pm.00}$ |
| EDM* Cornet et al. (2024) | 50 | $97.6_{\pm.0}$ | $77.6_{\pm.5}$ | $90.2_{\pm.2}$ | $89.2_{\pm.2}$ | $4.6_{\pm.1}$ | $1.7_{\pm.5}$ | $16.4_{\pm.2}$ | $0.61_{\pm.00}$ | $0.46_{\pm.00}$ |
| | 100 | $98.1_{\pm.0}$ | $81.9_{\pm.4}$ | $92.1_{\pm.2}$ | $90.9_{\pm.2}$ | $3.5_{\pm.1}$ | $1.4_{\pm.3}$ | $13.5_{\pm.1}$ | $0.61_{\pm.00}$ | $0.46_{\pm.00}$ |
| | 250 | $98.3_{\pm.0}$ | $84.3_{\pm.1}$ | $93.2_{\pm.4}$ | $91.7_{\pm.3}$ | $2.8_{\pm.2}$ | $1.3_{\pm.4}$ | $12.3_{\pm.4}$ | $0.62_{\pm.00}$ | $0.46_{\pm.00}$ |
| | 500 | $98.4_{\pm.0}$ | $85.2_{\pm.5}$ | $93.5_{\pm.2}$ | $92.2_{\pm.3}$ | $2.6_{\pm.2}$ | $1.3_{\pm.4}$ | $11.7_{\pm.1}$ | $0.62_{\pm.00}$ | $0.46_{\pm.00}$ |
| | 1000 | $98.4_{\pm.0}$ | $85.3_{\pm.3}$ | $93.5_{\pm.1}$ | $91.9_{\pm.1}$ | $2.5_{\pm.1}$ | $1.4_{\pm.4}$ | $11.3_{\pm.1}$ | $0.62_{\pm.00}$ | $0.46_{\pm.00}$ |
| GEOBFN Song et al. (2024) | 50 | $98.3_{\pm.1}$ | $85.1_{\pm.5}$ | $92.3_{\pm.4}$ | $90.7_{\pm.3}$ | – | – | – | – | – |
| | 100 | $98.6_{\pm.1}$ | $87.2_{\pm.3}$ | $93.0_{\pm.3}$ | $91.5_{\pm.3}$ | – | – | – | – | – |
| | 500 | $98.8_{\pm.8}$ | $88.4_{\pm.2}$ | $93.4_{\pm.2}$ | $91.8_{\pm.2}$ | – | – | – | – | – |
| | 1000 | $99.1_{\pm.1}$ | $90.9_{\pm.2}$ | $95.3_{\pm.1}$ | $93.0_{\pm.1}$ | – | – | – | – | – |
| MOTIFLOW | 50 | $99.1_{\pm.0}$ | $92.3_{\pm.3}$ | $96.6_{\pm.1}$ | $88.4_{\pm.2}$ | $2.41_{\pm.08}$ | $2.2_{\pm.3}$ | $13.6_{\pm.5}$ | $0.70_{\pm.00}$ | $0.47_{\pm.00}$ |
| | 100 | $99.1_{\pm.1}$ | $92.6_{\pm.5}$ | $95.3_{\pm.6}$ | $86.3_{\pm.9}$ | $2.24_{\pm.10}$ | $2.7_{\pm.4}$ | $10.3_{\pm.4}$ | $0.70_{\pm.00}$ | $0.47_{\pm.00}$ |

### A.4.2. WALL-CLOCK SAMPLING TIME

To assess how the compressed molecular representation and reduced sampling steps translate into inference speed, we track the wall-clock time required to sample 1024 molecules from QM9 in batches of size 128. All methods were evaluated on the same `GTX 1080` GPU. We sampled the molecular size from the marginal distribution, either at the atom level for the baselines or at the fragment level for MOTIFLOW. Crucially, the measured time for MOTIFLOW explicitly includes the reconstruction to the atom level, ensuring a fair comparison with atom-based methods. We compare the generation-step configurations reported in the respective papers. The results are provided in Table 7.

EDM (Hoogeboom et al., 2022) and GEOLDM (Xu et al., 2023), which are discrete-time diffusion models with fixed step configurations, perform comparably. In contrast, END (Cornet et al., 2024) incurs a higher computational cost due to its learnable forward process. Overall, MOTIFLOW achieves the lowest sampling time, with GEOBFN (Song et al., 2024) being the closest competitor. However, the performance gap widens for larger molecules: e.g., on GEOM-DRUGS, sampling from GEOBFN requires 213.3 and 437.9 seconds for 50 and 100 steps, respectively. This is more than double the time required by MOTIFLOW, which completes the task in 100.7 and 196.2 seconds, respectively. These results further highlight the efficiency of utilising a rigid-body motif representation for 3D molecular generation.

### A.4.3. CONDITIONAL GENERATION

Results on the atom composition and substructural features conditioning with additional generation steps configurations are reflected in Table 8. Figures 8 and 9 present instances of conditionally sampled molecules for both tasks.

*Table 6.* Extended results at unconditional generation on GEOM-Drugs. [†]Song et al. (2023) uses Dopri5 adaptive solver. Results obtained by Cornet et al. (2024) on the model of Xu et al. (2023) are denoted with [‡].

| Method | Steps | Stability A, % ($\uparrow$) | V $\times$ C % ($\uparrow$) | TV ($\downarrow$) A$[10^{-2}]$ | SA ($\uparrow$) | QED ($\uparrow$) |
|---|---|---|---|---|---|---|
| Data | – | 86.5 | 99.0 | – | 0.83 | 0.67 |
| EDM Hoogeboom et al. (2022) | 1000 | 81.3 | – | – | – | – |
| EDM-BRIDGE Wu et al. (2022) | 1000 | 82.4 | – | – | – | – |
| GEOLDM Xu et al. (2023) | 1000 | 84.4 | 45.8[‡] | 10.6 | – | – |
| EQUIFM Song et al. (2023) | –[†] | 84.1 | – | – | – | – |
| END Cornet et al. (2024) | 50 | $87.1_{\pm.1}$ | $66.0_{\pm.4}$ | $5.9_{\pm.1}$ | $0.62_{\pm.00}$ | $0.48_{\pm.00}$ |
| | 100 | $87.2_{\pm.1}$ | $73.7_{\pm.4}$ | $4.5_{\pm.1}$ | $0.66_{\pm.00}$ | $0.55_{\pm.00}$ |
| | 250 | $87.1_{\pm.1}$ | $77.4_{\pm.4}$ | $3.5_{\pm.0}$ | $0.69_{\pm.00}$ | $0.58_{\pm.00}$ |
| | 500 | $87.0_{\pm.0}$ | $78.6_{\pm.3}$ | $3.3_{\pm.0}$ | $0.70_{\pm.00}$ | $0.59_{\pm.00}$ |
| | 1000 | $87.0_{\pm.0}$ | $79.4_{\pm.4}$ | $3.0_{\pm.0}$ | $0.70_{\pm.00}$ | $0.59_{\pm.00}$ |
| EDM* Cornet et al. (2024) | 50 | $84.7_{\pm.0}$ | $46.6_{\pm.3}$ | $10.5_{\pm.1}$ | $0.59_{\pm.00}$ | $0.48_{\pm.00}$ |
| | 100 | $85.2_{\pm.1}$ | $56.2_{\pm.4}$ | $8.0_{\pm.1}$ | $0.61_{\pm.00}$ | $0.53_{\pm.00}$ |
| | 250 | $85.4_{\pm.0}$ | $61.4_{\pm.6}$ | $6.7_{\pm.1}$ | $0.63_{\pm.00}$ | $0.56_{\pm.00}$ |
| | 500 | $85.4_{\pm.0}$ | $63.4_{\pm.1}$ | $6.4_{\pm.1}$ | $0.64_{\pm.00}$ | $0.57_{\pm.00}$ |
| | 1000 | $85.3_{\pm.1}$ | $64.2_{\pm.6}$ | $6.2_{\pm.0}$ | $0.64_{\pm.00}$ | $0.57_{\pm.00}$ |
| GEOBFN Song et al. (2024) | 50 | 75.1 | – | – | – | – |
| | 100 | 78.9 | – | – | – | – |
| | 500 | 81.4 | – | – | – | – |
| | 1000 | 85.6 | – | – | – | – |
| MOTIFLOW | 50 | $96.2_{\pm.0}$ | $81.3_{\pm.2}$ | $8.1_{\pm.1}$ | $0.69_{\pm.00}$ | $0.72_{\pm.00}$ |
| | 100 | $96.3_{\pm.0}$ | $82.4_{\pm.2}$ | $7.4_{\pm.1}$ | $0.69_{\pm.00}$ | $0.72_{\pm.00}$ |

### A.4.4. ABLATION DETAILS

**Fragmentation statistics** In Section 5.3, we compare our proposed PLANAR RINGS decomposition strategy against a fine-grained NO RINGS variant and across different occurrence thresholds $\alpha$. Here, we provide detailed statistics on these decompositions for the training subsets of GEOM-DRUGS and QMUGS.

**Dataset properties**: the general statistics of the molecules in the training splits used for the ablation studies are summarised in Table 9. QMUGS contains generally larger and heavier molecules than GEOM-DRUGS.

**Fragmentation settings**: the NO RINGS strategy decomposes molecules by cutting all bonds except those to hydrogens and double/triple bonds, effectively breaking rings. For the PLANAR RINGS strategy, we vary the hyperparameter $\alpha$, reported as a percentage of the total dataset size. This threshold determines the minimum absolute occurrence count required for a motif to be retained in the vocabulary $\mathcal{V}_m$; motifs appearing less frequently are recursively decomposed.

Table 10 details the resulting fragment statistics. We observe that lower $\alpha$ thresholds (e.g., $0.01\%$) lead to a larger vocabulary with larger motifs up to 23 atoms and fewer fragments per molecule. Conversely, the NO RINGS baseline yields the highest number of fragments per molecule with the smallest maximum fragment size. Note that the fragment size includes necessary dummy atoms added to lock the coordinate frames of collinear or single-atom motifs. Across all configurations and datasets, the maximum size of the discrete symmetry group $|\mathcal{S}_i|$ encountered for any motif was 12.

We provide the extended results of our ablation study on the unconditional performance of different fragmentation methods in Table 11. The training on GEOM-DRUGS is done for 70 epochs, and for 50 epochs on QMUGS.

*Table 7.* Wall-clock time in seconds for sampling 1024 molecules on the QM9 dataset. $^{\dagger}$Song et al. (2023) use Dopri5 adaptive solver.

| Method | Steps | Time, s ($\downarrow$) |
|---|---|---|
| EDM Hoogeboom et al. (2022) | 1000 | 1875.9 |
| END Cornet et al. (2024) | 50 | 155.2 |
| | 100 | 304.1 |
| | 250 | 784.1 |
| | 500 | 1541.3 |
| | 1000 | 3156.3 |
| GEOLDM Xu et al. (2023) | 1000 | 1819.0 |
| EQUIFM Song et al. (2023) | $-^{\dagger}$ | 476.5 |
| GEOBFN Song et al. (2024) | 50 | 33.9 |
| | 100 | 68.0 |
| | 250 | 171.3 |
| | 500 | 357.4 |
| | 1000 | 685.5 |
| MOTIFLOW | 50 | 29.4 |
| | 100 | 56.1 |

**Architectural Ablations**    We ablate the design choices used in our main method presented in Section 3 on the unconditional generation with QM9. The results are summarised in Table 12.

**Discrete Prior**: we first evaluate the choice of the prior distribution for the discrete flow. While our default method uses a masking prior $m_0 = [\text{MASK}]$, the UNIFORM PRIOR variant employs a probability path that interpolates from a uniform categorical distribution over the vocabulary $\mathcal{V}_m$ to the target class.

**Motif Symmetry Handling**: to validate our dynamic alignment strategy, we compare it against three alternative approaches for handling motif symmetries. First, we consider a naive baseline NO $\mathcal{L}_{\text{GEODIFF}}$ that ignores symmetry, regressing directly towards the arbitrary orientation fixed during preprocessing. Second, in WITH $\mathcal{L}_{\text{AF3}}$, we test the alignment mechanism from ALPHAFOLD3 (Abramson et al., 2024), which aligns the target frame to the model's prediction $\hat{\mathbf{R}}_1$ rather than the noisy input $\mathbf{R}_t$. Third, instead of resolving symmetries analytically in the loss, we employ data augmentation in the WITH AUGMENTATION variant, where we randomly sample a symmetry element $\mathbf{S} \in \mathcal{S}_k$ from the motif's orbit at each training step and apply it to the target: $\tilde{\mathbf{R}}_1^k \leftarrow \mathbf{R}_1^k \mathbf{S}$.

**Backbone Components**: finally, we assess the contributions of specific architectural features. We evaluate the model performance without the triangle multiplicative updates in NO $\Delta$-UPDATE. Additionally, we train a variant with NO SELF-CONDITIONING to measure the impact of feeding the model's own discrete predictions of the clean motif types $\hat{m}_1$ back as input.

The results indicate that the most influential factor on the quality of unconditional generation is the removal of the triangle multiplicative update, confirming our hypothesis that it is useful for efficient modelling of unstructured molecular fragments. We did not observe a significant effect from applying data augmentation, since, strictly speaking, the target rotations are already chosen arbitrarily from equivalent orientations during preprocessing. Overall, our base method performs best, and thus reinforces the design choices outlined in Section 3.

*Table 8.* Results on atom composition and substructural features conditional generation tasks.

| Method | Steps | COMPOSITION | SUBSTRUCTURE |
|---|---|---|---|
| | | Matching, % ($\uparrow$) | Tanimoto Sim. ($\uparrow$) |
| CEDM Bao et al. (2023) | 1000 | – | $.671_{\pm.004}$ |
| EEGSDE Bao et al. (2023) | 1000 | – | $.750_{\pm.003}$ |
| CEDM* Cornet et al. (2024) | 50 | $69.6_{\pm0.6}$ | $.601_{\pm.000}$ |
| | 100 | $73.0_{\pm0.6}$ | $.640_{\pm.002}$ |
| | 250 | $74.1_{\pm1.4}$ | $.663_{\pm.002}$ |
| | 500 | $76.2_{\pm0.6}$ | $.669_{\pm.001}$ |
| | 1000 | $75.5_{\pm0.5}$ | $.673_{\pm.002}$ |
| CEND Cornet et al. (2024) | 50 | $89.2_{\pm0.8}$ | $.783_{\pm.001}$ |
| | 100 | $90.1_{\pm1.0}$ | $.807_{\pm.001}$ |
| | 250 | $91.2_{\pm0.8}$ | $.819_{\pm.001}$ |
| | 500 | $91.5_{\pm0.8}$ | $.825_{\pm.001}$ |
| | 1000 | $91.0_{\pm0.9}$ | $.828_{\pm.001}$ |
| MOTIFLOW | 50 | $92.2_{\pm0.4}$ | $.830_{\pm.001}$ |
| | 100 | $95.4_{\pm0.5}$ | $.862_{\pm.002}$ |

*Table 9.* Statistics of the training subsets of the datasets used in the ablation study.

| Dataset | Total Atoms | | | Heavy Atoms | | |
|---|---|---|---|---|---|---|
| | Mean | Median | Max | Mean | Median | Max |
| GEOM-DRUGS | 45.2 | 45.0 | 143 | 25.3 | 25.0 | 77 |
| QMUGS | 53.6 | 52.0 | 178 | 29.9 | 29.0 | 96 |

*Table 10.* Detailed statistics of the ablated rigid-motif decomposition strategies. The *Cut-off* column indicates the absolute number of occurrences required for a motif to be preserved in the vocabulary. *Max. size* denotes the maximum number of atoms, including dummy ones, in a single motif.

| Dataset | Fragmentation | Cut-off | Fragments per molecule | | | Max. size |
|---|---|---|---|---|---|---|
| | | | Mean | Median | Max | |
| GEOM-DRUGS | NO RINGS | – | 16.8 | 17.0 | 57 | 6 |
| | PLANAR RINGS 0.5% | 964 | 13.7 | 13.0 | 57 | 17 |
| | PLANAR RINGS 0.1% | 192 | 13.1 | 13.0 | 57 | 17 |
| | PLANAR RINGS 0.01% | 19 | 12.7 | 12.0 | 57 | 23 |
| QMUGS | NO RINGS | – | 19.9 | 19.0 | 80 | 6 |
| | PLANAR RINGS 0.5% | 1419 | 16.7 | 16.0 | 78 | 17 |
| | PLANAR RINGS 0.1% | 283 | 15.9 | 15.0 | 78 | 17 |
| | PLANAR RINGS 0.01% | 28 | 15.3 | 15.0 | 78 | 23 |

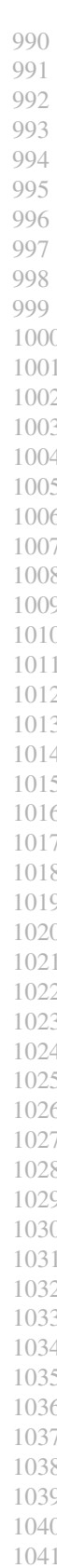

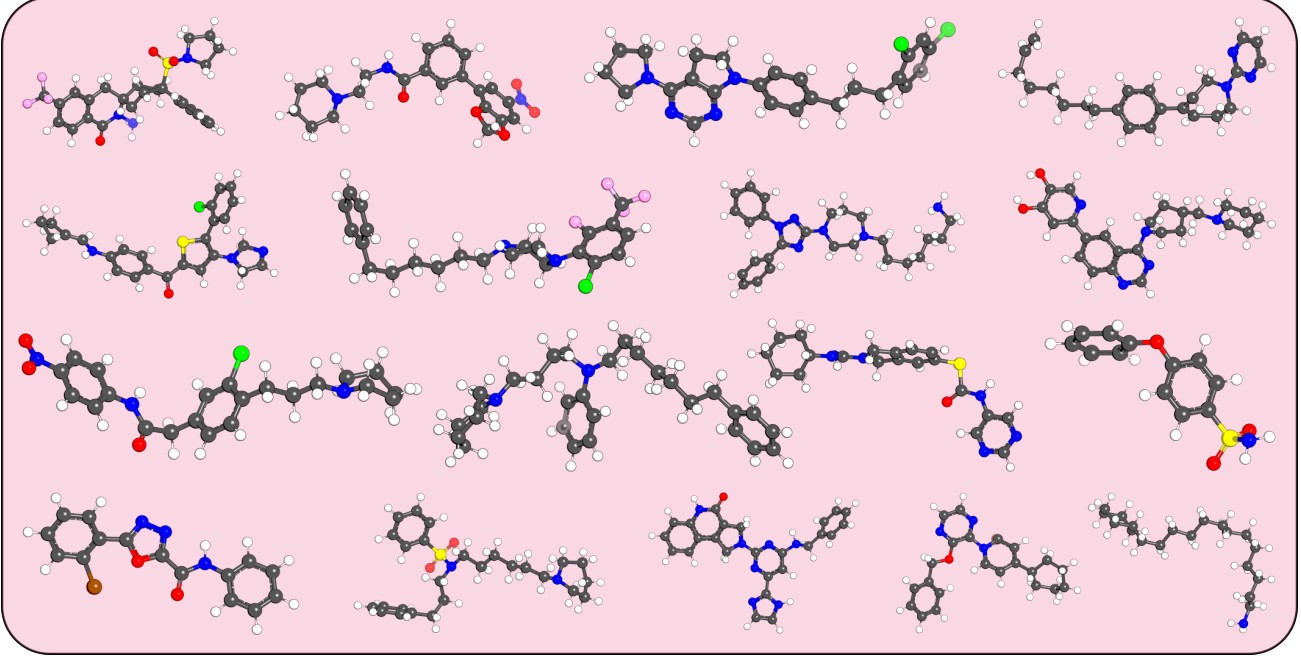

*Figure 6.* Randomly selected molecules that are sampled unconditionally from MOTIFLOW trained on GEOM-DRUGS.

*Figure 7.* Randomly selected molecules that are sampled unconditionally from MOTIFLOW trained on QMUGS.

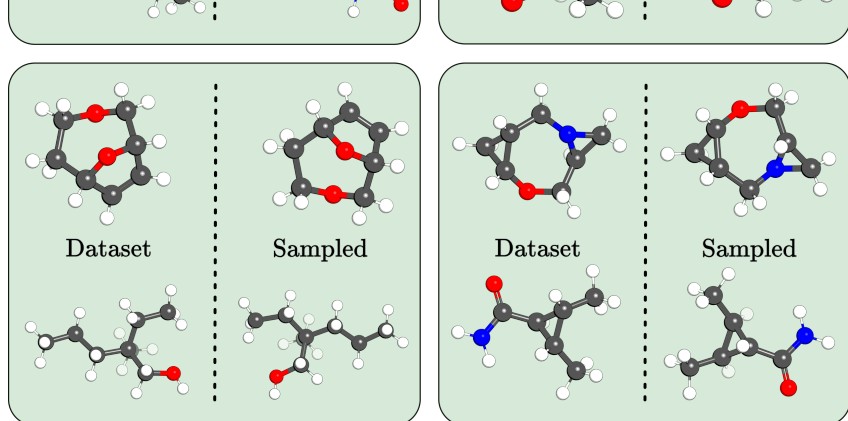

*Figure 8.* Randomly chosen samples from the atom composition conditional task.

*Figure 9.* Randomly chosen samples from the fingerprint substructure conditional task.

*Table 11.* Extended ablation results for the unconditional generation on GEOM-DRUGS and QMUGS for various fragmentation strategies.

| Fragmentation | Steps | QMUGS | | | | | GEOM-DRUGS | | | | |
|---|---|---|---|---|---|---|---|---|---|---|---|
| | | Stability A, % (↑) | V × C % (↑) | SA (↑) | QED (↑) | TV (↑) A $[10^{-2}]$ | Stability A, % (↑) | V × C % (↑) | SA (↑) | QED (↑) | TV (↑) A $[10^{-2}]$ |
| No Rings | 50 | $86.5_{\pm0.6}$ | $83.0_{\pm.2}$ | $0.74_{\pm.00}$ | $0.51_{\pm.00}$ | $5.12_{\pm.06}$ | $89.1_{\pm0.7}$ | $80.9_{\pm.2}$ | $0.71_{\pm.00}$ | $0.69_{\pm.00}$ | $3.57_{\pm.05}$ |
| | 100 | $85.5_{\pm1.8}$ | $85.6_{\pm.3}$ | $0.74_{\pm.00}$ | $0.50_{\pm.00}$ | $4.98_{\pm.06}$ | $86.7_{\pm1.8}$ | $82.9_{\pm.4}$ | $0.71_{\pm.00}$ | $0.69_{\pm.00}$ | $3.36_{\pm.08}$ |
| Planar Rings 0.5% | 50 | $95.1_{\pm0.1}$ | $82.2_{\pm.8}$ | $0.72_{\pm.00}$ | $0.60_{\pm.00}$ | $8.56_{\pm.04}$ | $94.9_{\pm0.0}$ | $81.3_{\pm.4}$ | $0.70_{\pm.00}$ | $0.71_{\pm.00}$ | $4.47_{\pm.02}$ |
| | 100 | $95.5_{\pm0.1}$ | $84.0_{\pm.5}$ | $0.72_{\pm.00}$ | $0.60_{\pm.00}$ | $8.08_{\pm.17}$ | $95.2_{\pm0.1}$ | $82.1_{\pm.3}$ | $0.70_{\pm.00}$ | $0.71_{\pm.00}$ | $4.24_{\pm.14}$ |
| Planar Rings 0.1% | 50 | $95.7_{\pm0.0}$ | $81.6_{\pm.3}$ | $0.72_{\pm.00}$ | $0.58_{\pm.00}$ | $8.47_{\pm.08}$ | $94.8_{\pm0.0}$ | $79.6_{\pm.5}$ | $0.70_{\pm.00}$ | $0.68_{\pm.00}$ | $4.45_{\pm.12}$ |
| | 100 | $96.1_{\pm0.0}$ | $83.1_{\pm.4}$ | $0.72_{\pm.00}$ | $0.58_{\pm.00}$ | $8.01_{\pm.22}$ | $95.0_{\pm0.0}$ | $81.2_{\pm.3}$ | $0.69_{\pm.00}$ | $0.68_{\pm.00}$ | $3.93_{\pm.09}$ |
| Planar Rings 0.01% | 50 | $95.1_{\pm0.0}$ | $78.8_{\pm.3}$ | $0.72_{\pm.00}$ | $0.49_{\pm.01}$ | $11.62_{\pm.11}$ | $96.2_{\pm0.0}$ | $81.3_{\pm.2}$ | $0.69_{\pm.00}$ | $0.72_{\pm.00}$ | $8.07_{\pm.06}$ |
| | 100 | $95.3_{\pm0.0}$ | $80.4_{\pm.5}$ | $0.73_{\pm.00}$ | $0.49_{\pm.00}$ | $11.10_{\pm.03}$ | $96.3_{\pm0.0}$ | $82.4_{\pm.3}$ | $0.69_{\pm.00}$ | $0.72_{\pm.00}$ | $7.43_{\pm.05}$ |

*Table 12.* Architectural ablations on QM9.

| Method | Steps | Stability (↑) | | Val. / Uniq. (↑) | | TV (↓) | | Str. En. (↓) | SA (↑) | QED (↑) |
|---|---|---|---|---|---|---|---|---|---|---|
| | | A, % | M, % | V, % | V × U, % | A $[10^{-2}]$ | B $[10^{-3}]$ | ΔE $\left[\frac{kcal}{mol}\right]$ | | |
| Data | – | 99.0 | 95.2 | 97.7 | 97.7 | – | – | 7.7 | 0.63 | 0.46 |
| Uniform Prior | 50 | $98.7_{\pm.1}$ | $90.8_{\pm.1}$ | $95.2_{\pm.5}$ | $89.2_{\pm.5}$ | $1.42_{\pm.06}$ | $12.2_{\pm.6}$ | $14.3_{\pm.5}$ | $0.70_{\pm.00}$ | $0.47_{\pm.00}$ |
| | 100 | $98.7_{\pm.1}$ | $90.4_{\pm.4}$ | $93.5_{\pm.3}$ | $87.5_{\pm.4}$ | $1.38_{\pm.09}$ | $15.2_{\pm.2}$ | $10.7_{\pm.1}$ | $0.70_{\pm.00}$ | $0.47_{\pm.00}$ |
| No $\mathcal{L}_{\text{GeoDiff}}$ | 50 | $98.6_{\pm.0}$ | $89.5_{\pm.2}$ | $95.5_{\pm.1}$ | $88.7_{\pm.5}$ | $2.35_{\pm.05}$ | $4.9_{\pm.5}$ | $16.5_{\pm.2}$ | $0.70_{\pm.00}$ | $0.47_{\pm.00}$ |
| | 100 | $98.9_{\pm.0}$ | $90.8_{\pm.2}$ | $94.5_{\pm.1}$ | $88.0_{\pm.4}$ | $2.35_{\pm.08}$ | $6.4_{\pm.5}$ | $11.8_{\pm.1}$ | $0.70_{\pm.00}$ | $0.47_{\pm.00}$ |
| With $\mathcal{L}_{\text{AF3}}$ | 50 | $98.5_{\pm.0}$ | $88.7_{\pm.2}$ | $93.8_{\pm.2}$ | $85.4_{\pm.4}$ | $2.81_{\pm.04}$ | $4.0_{\pm.7}$ | $16.6_{\pm.6}$ | $0.70_{\pm.00}$ | $0.47_{\pm.00}$ |
| | 100 | $98.6_{\pm.1}$ | $89.4_{\pm.5}$ | $92.4_{\pm.5}$ | $83.6_{\pm.4}$ | $2.87_{\pm.05}$ | $5.6_{\pm1.5}$ | $12.1_{\pm.5}$ | $0.71_{\pm.00}$ | $0.47_{\pm.00}$ |
| With Augmentation | 50 | $98.8_{\pm.0}$ | $90.8_{\pm.3}$ | $95.9_{\pm.1}$ | $87.9_{\pm.5}$ | $2.44_{\pm.05}$ | $4.6_{\pm.8}$ | $15.4_{\pm.5}$ | $0.70_{\pm.00}$ | $0.47_{\pm.00}$ |
| | 100 | $98.8_{\pm.0}$ | $90.2_{\pm.3}$ | $93.5_{\pm.3}$ | $85.0_{\pm.4}$ | $2.32_{\pm.07}$ | $8.4_{\pm1.3}$ | $11.3_{\pm.3}$ | $0.70_{\pm.00}$ | $0.47_{\pm.00}$ |
| No Δ-Update | 50 | $98.3_{\pm.1}$ | $86.2_{\pm.6}$ | $93.4_{\pm.4}$ | $85.7_{\pm.5}$ | $2.28_{\pm.08}$ | $4.4_{\pm.5}$ | $16.9_{\pm.5}$ | $0.70_{\pm.00}$ | $0.47_{\pm.00}$ |
| | 100 | $98.2_{\pm.1}$ | $85.7_{\pm1.4}$ | $90.8_{\pm1.4}$ | $82.0_{\pm1.3}$ | $2.51_{\pm.08}$ | $6.8_{\pm1.8}$ | $13.3_{\pm.1}$ | $0.70_{\pm.00}$ | $0.47_{\pm.00}$ |
| No Self-Conditioning | 50 | $98.9_{\pm.0}$ | $91.0_{\pm.3}$ | $96.1_{\pm.3}$ | $88.3_{\pm.4}$ | $1.73_{\pm.05}$ | $4.7_{\pm.4}$ | $15.5_{\pm.3}$ | $0.70_{\pm.00}$ | $0.47_{\pm.00}$ |
| | 100 | $98.8_{\pm.0}$ | $90.6_{\pm.3}$ | $93.7_{\pm.3}$ | $85.2_{\pm.5}$ | $1.66_{\pm.04}$ | $7.4_{\pm.1}$ | $10.9_{\pm.2}$ | $0.70_{\pm.00}$ | $0.47_{\pm.00}$ |
| MotiFlow | 50 | $99.1_{\pm.0}$ | $92.3_{\pm.3}$ | $96.6_{\pm.1}$ | $88.4_{\pm.2}$ | $2.41_{\pm.08}$ | $2.2_{\pm.3}$ | $13.6_{\pm.5}$ | $0.70_{\pm.00}$ | $0.47_{\pm.00}$ |
| | 100 | $99.1_{\pm.1}$ | $92.6_{\pm.5}$ | $95.3_{\pm.6}$ | $86.3_{\pm.9}$ | $2.24_{\pm.10}$ | $2.7_{\pm.4}$ | $10.3_{\pm.4}$ | $0.70_{\pm.00}$ | $0.47_{\pm.00}$ |

