# OpenReview forum: "3D Molecule Generation from Rigid Motifs via $\mathrm{SE}(3)$ Flows"
_ICML.cc/2026/Conference — Submitted to ICML 2026_

### Official Review · Reviewer_u7tZ · 2026-03-07

**Soundness:** 3
**Presentation:** 3
**Significance:** 2
**Originality:** 2
**Overall Recommendation:** 4
**Confidence:** 3

**Summary:**

This paper improves the existing molecular generation frameworks in the following aspects:

- It replaces the direct generation of 3D coordinates with the generation of rigid motifs.
- It adopts the more advanced discrete flow matching models to generate the discrete features.
- It improves the modeling of SE(3) from diffusion models to flow matching models.

The proposed framework achieves strong experimental results in the existing molecular generation baselines.

**Compliance With Llm Reviewing Policy:**

Affirmed.

**Final Justification:**

I thank the authors for fully addressing each of the points I have raised. However, as these concerns have not substantially affected my overall assessment of the manuscript, I have decided to maintain my original scores.

**Key Questions For Authors:**

As mentioned in the weaknesses, the proposed framework introduces three main modifications to existing baselines. However, the current ablation study only focuses on the motif fragmentation strategies. It remains unclear how much of the performance gain is attributed to each of the three architectural changes. Could the authors provide a system-level ablation study to isolate the effect of these three components?

**Limitations:**

yes

**Strengths And Weaknesses:**

# Strengths
- The proposed method is well-motivated. There is plenty of literature supporting the effectiveness of these adaptations. For instance, modeling on rigid motifs has been proven effective in SE(3) diffusion model with application to protein backbone generation and Torsional Diffusion (a relevant missing reference in this paper). Discrete flow matching is a recent technique whose application to other domains has proven highly effective, and the transition from diffusion to flow matching is a natural improvement.
- The proposed method demonstrates strong empirical performance on the QM9 and the Drugs dataset, both in the sampling quality and sampling efficiency.
- The authors provide a detailed ablation on fragmentation, demonstrating the trade-off between the vocabulary size and the generation quality.

# Weaknesses
- The proposed method is a direct combination of several existing methods. The performance gain seems to stem directly from these basic modeling frameworks. Hence, the contribution feels somewhat incremental; it reads more like assembling existing techniques rather than proposing a fundamentally new mechanism, despite the strong experimental results.

Minors. Fig. 5 appears to be made from a screenshot or png picture, which is not suitable for a conference paper.

I am familiar with the methods used in this paper, but not very familiar with the experimental metrics.

---

> ### Author Rebuttal · Authors · 2026-03-31
>
> We sincerely thank the reviewer for their time, constructive feedback, highlighting the strong empirical performance and detailed fragmentation ablations of our work, and assessing our contribution as well-motivated. We address their remaining points and questions below.
>
> > Q1: System-level ablation to isolate the three architectural components.
>
> We would like to clarify that these ablations were already performed and are reported in **Table 12 (Appendix A.4.4)**, with a pointer at the end of the Method section. We systematically isolated all key architectural design choices, including: (i) the prior for the discrete flow (fully masked vs. uniform); (ii) the handling of motif rotation automorphisms (GeoDiff-style alignment vs. AlphaFold3-style vs. data augmentation); and (iii) the IPA backbone modifications (removal of triangle multiplicative updates; removal of self-conditioning). The results show that the most influential factor is the triangle multiplicative update, confirming its utility for modelling unstructured rigid molecular fragments (as opposed to linear protein backbones). Our base configuration performs best overall. We apologise if the placement in the appendix made these results difficult to find, and we will add a more prominent pointer in the main text.
>
> > Q2: The contribution feels incremental; assembling existing techniques rather than proposing a fundamentally new mechanism.
>
> We fully agree that our method combines existing modelling techniques rather than introducing a new generative paradigm, and we do not claim otherwise. Our core contribution is a fundamental shift in **how 3D molecules are represented** during generation: we change the foundational building blocks from atoms to rigid motifs. While motif-based generation is well-explored for abstract 2D molecular graphs, and rigid-frame residue generation is standard for proteins (with a fixed, small vocabulary and linear topologies), bridging these ideas to *de novo* 3D generation of general drug-like molecules is highly non-trivial. Small molecules exhibit arbitrary branching, diverse spatial topologies, a vastly larger motif vocabulary with varied automorphisms, and require careful handling of rotatable bonds, as well as fundamentally different fragmentation compared to molecular graph methods. Formulating the generative task on the SE(3) manifold of fragments to address these specific challenges is, we believe, a non-incremental contribution to 3D drug-like molecule design.
>
> > Q3: "The performance gain seems to stem directly from the advanced modelling frameworks."
>
> We respectfully argue that our ablations demonstrate otherwise: the performance leap stems directly from the **rigid-motif representation**, and this can be isolated while holding the architecture and flow matching framework entirely constant.
>
> As shown in Table 4, the **No Rings** configuration, which preserves only hydrogen bonds and double/triple bonds, breaking all rings, serves as the closest conceptual proxy to atom-level modelling that our SE(3) framework can accommodate (since it was designed for rigid frames, not individual atoms in $\mathbb{R}^3$).
>
> Transitioning from this fine-grained No Rings proxy to the **Planar Rings 0.5%** configuration (which retains some rigid ring structures as single entities) produces a notable improvement in atom stability: **86.7% → 95.2%** on GEOM-Drugs and **85.5% → 95.5%** on QMugs; a ~10 percentage point jump from preserving some rings alone.
>
> This cleanly isolates the value of the proposed representation: distorted planar geometry and broken rings are well-documented failure modes of atom-level flow-based models (e.g., [1]), because denoising individual atoms into rigid cycles is geometrically ill-conditioned. By treating these structures as rigid motifs, MotiFlow bypasses this failure mode entirely. This is why it outperforms advanced atom-based baselines such as flow-based EquiFM and methods spanning latent diffusion to Bayesian flow networks.
>
> > Minor points
>
> * Torsional Diffusion for Molecular Conformer Generation (Jing et al., 2022): We thank the reviewer for this pointer. While it focuses on conformation generation over torsional angles of a known graph rather than *de novo* SE(3) generation, it shares the philosophy of leveraging rigid substructures to constrain the generation space. We will add the reference and a brief discussion to the Rigid-body Generation paragraph in Related Work.
> * Figure 5: We thank the reviewer for flagging the format issue. We are unable to modify the manuscript during the rebuttal period, but we will replace it with a high-quality vector graphic in the camera-ready version.
>
> We want to thank the reviewer again for their effort in reviewing and improving our manuscript. We hope this addresses all concerns and welcome further discussion.
>
> [1]: MolDiff: Addressing the Atom-Bond Inconsistency Problem in 3D Molecule Diffusion Generation (Peng et al., 2023)

---

> > ### Author Rebuttal · Reviewer_u7tZ · 2026-04-01
> >
> > I thank the authors for fully addressing each of the points that I have raised, and I have decided to maintain my score.

---

### Official Review · Reviewer_fQUj · 2026-03-11

**Soundness:** 2
**Presentation:** 3
**Significance:** 2
**Originality:** 2
**Overall Recommendation:** 3
**Confidence:** 3

**Summary:**

This paper proposes MOTIFLOW, a framework that represents druglike small molecules as sets of rigid motifs on the SE(3) manifold. It integrates SE(3)-equivariant flow matching and discrete flow modeling for 3D molecule generation.

**Compliance With Llm Reviewing Policy:**

Affirmed.

**Key Questions For Authors:**

The performance gain on the QM9 dataset is insignificant, and uniqueness even decreases. Please provide a clear explanation and address whether motif-level representation has inherent drawbacks for small molecules.

**Limitations:**

The performance gain is insignificant on the small-molecule dataset QM9, and some metrics decrease, indicating that motif-level representation has an inherent bias on small systems and is not optimal for all molecular sizes.

**Strengths And Weaknesses:**

Strengths:
The method constructs molecular representations via a datadriven rigidmotif decomposition and canonicalization strategy, jointly learning the distribution of motif types and their spatial geometric configurations. It achieves or surpasses stateoftheart performance on benchmarks including GEOMDRUGS, QM9, and QMUGS, with significantly improved atom stability, 2–10× fewer generation steps, 3.5× compression in molecular representations, and efficient unconditional and conditional generation.

Weaknesses:
1) Although the paper claims to systematically transfer the SE(3) rigid-frame generation paradigm from protein backbones to general small-molecule 3D generation for the first time, SE(3)-based methods have been extensively studied in material generation and prediction. It remains unclear whether the paper has introduced sufficient task-specific adaptations and genuine innovations beyond conceptual reuse.
2) The method relies on a predefined motif vocabulary, and there exists an inherent trade-off between vocabulary size and generalization capability.
3) The proposed method only requires 100 sampling steps to outperform 1000-step diffusion models, achieving a one-order-of-magnitude speedup and drastically reduced inference time. However, does its performance advantage still hold when compared to other one-step diffusion methods?
4) Some equations are missing punctuation marks at the end.
5) The performance advantage of the proposed method on QM9 is not significant. Could the authors provide more explanations for this?

---

> ### Author Rebuttal · Authors · 2026-03-31
>
> We thank the reviewer for their efforts in reviewing our manuscript. Below, we address the concerns and questions raised point by point.
>
> > “The performance gain on the QM9 dataset is insignificant, and uniqueness even decreases. Please provide a clear explanation and address whether motif-level representation has inherent drawbacks for small molecules.”
>
> **Performance gain on QM9:** As established in recent literature [1, 2], unconditional 3D generation on QM9 is largely saturated, with atom stability already exceeding 98.5% for baselines, making it non-informative for distinguishing model performance [2]. The most rigorous and challenging standard metric remaining on this dataset is *molecular stability* (requiring *every* atom to be stable), on which MotiFlow achieves **92.6\%**, surpassing all baselines.
>
> **Uniqueness drop:** The drop in $V \times U$ stems from the sampling temperature in the discrete flow. We ran additional experiments varying the temperature annealing schedule (linear decrease $T_{start} \to T_{end}$), controlling the diversity-stability trade-off:
>
> | Temperature Schedule | Atom Stable, % | Molecule Stable, % | Valid $V$, % | $V \times U$, % |
> | :--- | :--- | :--- | :--- | :--- |
> | $1.0$ (Constant) | $99.3 \pm .1$ | $94.2 \pm .7$ | $96.3 \pm .4$ | $87.2 \pm .2$ |
> | $1.5 \to 1.0$ (Original) | $99.1 \pm .1$ | $92.6 \pm .5$ | $95.3 \pm .6$ | $86.3 \pm .9$ |
> | $2.0 \to 1.0$ | $99.0 \pm .0$ | **$91.4 \pm .3$** | $94.6 \pm .3$ | **$91.2 \pm .1$** |
> | $3.0 \to 1.0$ | $98.5 \pm .0$ | $87.5 \pm .3$ | $92.7 \pm .5$ | $90.4 \pm .7$ |
>
> With a schedule of $2.0 \to 1.0$, MotiFlow achieves a competitive $V \times U$ of 91.2%, while maintaining molecular stability of 91.4%, still surpassing all baselines. Since uniqueness is recoverable without sacrificing our lead in molecular stability, we respectfully argue there is no inherent bias for small molecules. The benefits of MotiFlow (conciseness, speedup, ability to resolve complex topologies) naturally shine brightest on larger molecules like those in GEOM-Drugs, which is precisely our primary use case.
>
> > “…unclear whether the paper has introduced sufficient task-specific adaptations and genuine innovations beyond conceptual reuse.”
>
> We provide a detailed response to an identical question in our rebuttal to Reviewer **u7tZ** (Q2), and kindly ask the reviewer to refer to it. In short: our core contribution is a fundamental shift in *how 3D molecules are represented during generation*, from atoms to rigid motifs. Bridging motif-based 2D graph generation and rigid-frame protein generation to *de novo* 3D generation of general drug-like molecules (with arbitrary branching, diverse topologies, a vastly larger motif vocabulary with varied automorphisms, requiring novel fragmentation) is non-trivial, and we believe it is a non-incremental contribution to 3D drug-like molecule design.
>
> > “…The method relies on a predefined motif vocabulary, and there exists an inherent trade-off between vocabulary size and generalisation capability.”
>
> We fully acknowledge this limitation and dedicated **Section 5.3** and **Appendix A.4.4** entirely to analysing it. This trade-off is a fundamental characteristic of *any* fragment-based generative model. Our work demonstrates, for the first time, how to harness the established advantages of motif-based generation (representational efficiency, chemical validity, modularity) in the 3D domain.
>
> > “…Does its performance advantage still hold when compared to other one-step diffusion methods?”
>
> Speed-up techniques for flow-based models (distillation, consistency models, etc.) are *orthogonal* to MotiFlow, since our improvements derive strictly from the **data representation**, not specialised acceleration tools. Crucially, this means MotiFlow is **fully compatible with such techniques**: it is built on standard SE(3) and categorical flow matching formulations, and can be augmented with distillation to achieve one-step generation. The most direct correspondence would combine [3] for motif-class generation with [4] for SE(3) modelling. Our primary contribution (the shift from atom space to rigid-motif space) is agnostic to acceleration methods and can be freely coupled with them, potentially yielding further gains beyond those reported.
>
> > Punctuation
>
> We have identified the missing full stop in the first equation of Section 3 (line 126) and will correct it in the revision.
>
> We hope these clarifications and an additional experiment address the reviewer's concerns, and we kindly ask the reviewer to consider revising their score in light of them.
>
> [1]: Applications of Modular Co-Design for De Novo 3D Molecule Generation (Reidenbach et al., 2025)
>
> [2]: SemlaFlow -- Efficient 3D Molecular Generation with Latent Attention and Equivariant Flow Matching (Irwin et al., 2025)
>
> [3]: Categorical Flow Maps (Roos et al., 2026)
>
> [4]: Generalised Flow Maps for Few-Step Generative Modelling on Riemannian Manifolds (Davis et al., 2026)

---

> > ### Author Rebuttal · Reviewer_fQUj · 2026-04-02
> >
> > Limited performance on small molecules.

---

> > > ### Author Response · Authors · 2026-04-03
> > >
> > > We thank the reviewer for their follow-up and wish to clarify some key points that may not have been fully communicated in our previous response.
> > >
> > > Regarding **QM9 performance**, our results are strong: MotiFlow achieves the highest molecular stability among all baselines ($92.6$%, or $91.4$% at the temperature setting that also restores $V \times U$ to a competitive $91.2$%). We appreciate the reviewer’s observation regarding the modest *gains* relative to baselines and understand this perspective. On a benchmark widely recognised as saturated (atom stability $\geq98.7$% across all methods), substantial improvements are structurally unattainable for any method. We view any improvement on the most challenging remaining metric (molecular stability) as meaningful. We note that our empirical results have also been referred to as “substantial” and “strong” by other reviewers.
> > >
> > > Regarding the **scope** of our work, our primary motivation is to generate *realistic, drug-like molecules* such as those found in GEOM-Drugs and QMugs, which include large rings and fused-ring systems. In this context, denoising individual atoms into geometrically rigid structures is ill-conditioned, as noted by [1]. While atom-based methods show limitations in scaling and performance in these settings, the rigid-motif abstraction offers significant advantages. QM9, which consists of small organic molecules with at most nine heavy atoms, is included in our study for completeness and comparability with prior work, rather than as a central claim. We hope this clarifies our intent and the context of our results.
> > >
> > > From a **theoretical perspective on small molecules**, we hope to address the reviewer’s concern directly. In the limit of maximal fragmentation, where each atom is treated as its own motif (which is achievable in our framework by introducing dummy atoms to lock orientations for isolated atoms and provide an inductive bias regarding their neighborhood), our method formally reduces to an “atom-level SE(3) flow”. Thus, our framework strictly generalises atom-based generation. The primary consideration is whether coarser motifs offer additional benefits. For QM9-scale molecules, these benefits are modest because their small size allows atom-level methods to perform near-optimally, leaving little room for improvement. In contrast, for GEOM-Drugs and QMugs, the benefits are substantial, as demonstrated by our experimental results.
> > >
> > > Regarding **flexibility**, our framework provides practitioners with explicit control: they may select finer fragmentation (yielding higher uniqueness and atom-level expressivity) or coarser fragmentation (resulting in higher stability and greater efficiency) depending on the downstream application. We believe this flexibility in representations tailored to practitioner needs and application context is an advantage over atom-based methods.
> > >
> > > **Proposed revision**: We are happy to add a paragraph to the manuscript to make this scope explicit: the rigid-motif representation is most beneficial for medium- to large-sized drug-like molecules, where rings and fused-ring systems are prevalent. For small molecules such as those in QM9, the representation remains valid and competitive, but the gains are limited precisely because atom-level methods already perform near-optimally at that scale. We believe this framing accurately reflects the experimental evidence and sets appropriate expectations for readers.
> > >
> > > We thank the reviewer once again for their follow-up and hope that these clarifications address the remaining concern.
> > >
> > > [1]: MolDiff: Addressing the Atom-Bond Inconsistency Problem in 3D Molecule Diffusion Generation (Peng et al., 2023)

---

### Official Review · Reviewer_2gyL · 2026-03-13

**Soundness:** 2
**Presentation:** 3
**Significance:** 3
**Originality:** 2
**Overall Recommendation:** 5
**Confidence:** 3

**Summary:**

This paper introduces MOTI-FLOW, a novel generative framework for 3D molecular structure generation that operates on rigid-body motifs rather than individual atoms. The method demonstrates comparable or superior performance to state-of-the-art atom-based methods on QM9, GEOM-DRUGSbenchmarks, with 2×-10× fewer generation steps and 3.5× compression in molecular representations.

**Compliance With Llm Reviewing Policy:**

Affirmed.

**Final Justification:**

The authors addressed my main concerns in the rebuttal, and I maintain my score of 5 unchanged.

**Key Questions For Authors:**

See weaknesses.

**Limitations:**

yes

**Strengths And Weaknesses:**

# Strengths

1. Motif-based generation holds significant value for complex molecular systems, given that many substructural fragments within molecules are inherently fixed or conserved.
2. The authors have demonstrated considerable rigor and attention to detail in motif decomposition and representation.
3. The experimental improvements are substantial.

# Weaknesses

1. Does $V_m$ include individual atoms? What is the distinction between its representation and that of fragments?
2. A rotation matrix $R$ is employed within the motif to represent orientation. On what coordinate system is this based? If the global coordinate system undergoes rotation, does the model maintain its generalization capability for molecules within the dataset?
3. In the GEOM dataset, molecular generation often fails to yield complete molecules (e.g., generating multiple valid fragments that RDKit still identifies as a valid molecule). Can MOTI-FLOW help mitigate this issue?
4. The high validity rate achieved via motifs may be an expected outcome, as it incorporates priors regarding plausible structures. However, I have a concern: in conditional generation based on specified chemical properties, the generated results might deviate significantly from the target conditions. This is because MOTI-FLOW may substantially alter the molecular structure, making it difficult to navigate the specific property space defined by the given conditions.

---

> ### Author Rebuttal · Authors · 2026-03-31
>
> We thank the reviewer for their time and effort in reviewing our manuscript, and for highlighting the relevance, rigour, and empirical performance of our work. We address all questions and concerns below.
>
> > 1. “Does $V_m$ include individual atoms? What is the distinction between its representation and that of fragments?”
>
> Indeed, $V_m$ includes individual atoms. If the fragmentation algorithm (Section 3.1) produces a single-atom fragment, we add two dummy atoms placed at a unit distance to the two nearest non-collinear neighbours of this atom in 3D space. This defines a well-posed orientation $\mathbf{R} \in \mathrm{SO}(3)$ with an inductive bias for the atom's neighbourhood. The single-atom fragment thus has a well-posed rigid frame $\mathbf{T} \in \mathrm{SE}(3)$ (dummy atoms are used solely to lock orientation), and its representation is identical to that of any other fragment.
>
> > 2.1. "A rotation matrix $R$ is employed within the motif to represent orientation. On what coordinate system is this based?"
>
> TL;DR: a rotation applied to the centred coordinates of a canonical fragment chosen arbitrarily from the dataset and fixed.
>
> For a motif type $m$ with $N$ atoms, we pick one dataset instantiation $i$ as **canonical**: its centred coordinates $\mathbf{P}_i \in \mathbb{R}^{N \times 3}$ define the canonical pose. By rigidity, any further instantiation $j$ of the same type is described by rotating and translating $\mathbf{P}_i$. Let $\mathbf{Y}_j$ be its centred coordinates (same atom order up to equivalent permutations). The rotation $\mathbf{R}_j$ mapping $\mathbf{P}_i$ to $\mathbf{Y}_j$ is the motif's rotation matrix, computed via the Kabsch algorithm.
>
> > 2.2. "If the global coordinate system undergoes rotation, does the model maintain its generalization capability for molecules within the dataset?"
>
> Regarding global rotations, let's apply an arbitrary rotation $r \in \mathrm{SO}(3)$ to the coordinate system of a motif $j$: $\mathbf{Y}_j \cdot r$. Since we can express the pose of the motif $j$ using its frame $\mathbf{T}_j = (\mathbf{R}_j, \mathbf{x}_j)$ relative to the fixed canonical pose $\mathbf{P}_i$, we have:
> $\mathbf{Y}_j \cdot r = (\mathbf{P}_i \cdot \mathbf{R}_j + \mathbf{x}_j) \cdot r = \mathbf{P}_i \cdot (\mathbf{R}_j \cdot r) + (\mathbf{x}_j \cdot r)$.
> This means that rotating the global coordinate system is exactly equivalent to rotating the frame $\mathbf{T}_j$ by the same $r$. Because the IPA-based network $f(\cdot)$ we use is $\mathrm{SE}(3)$-equivariant, it follows that $f(\mathbf{T}_j \odot r) = f(\mathbf{T}_j) \odot r$. Therefore, MotiFlow is fully $\mathrm{SE}(3)$-equivariant, meaning any rotation of the global coordinate system at the input simply rotates the output identically, maintaining the model’s generalisation.
>
> > 3. "In the GEOM dataset, molecular generation often fails to yield complete molecules (e.g., generating multiple valid fragments that RDKit still identifies as a valid molecule). Can MOTI-FLOW help mitigate this issue?"
>
> The reviewer rightfully highlights this known issue with the RDKit’s validity metric, which does not enforce connectivity. In order to rigorously evaluate the quality of generation, we follow previous works (e.g., [1]) and estimate the percentage of **valid and connected** ($V \times C$; i.e., valid molecules that form strictly one connected component) generated molecules. As Table 2 shows, MotiFlow surpasses the closest baseline by 7.5% in this $V \times C$ metric, indicating a superior capability in generating both valid and complete (i.e., connected) molecules.
>
> > 4. "[...] generated results might deviate significantly from the target conditions. This is because MOTI-FLOW may substantially alter the molecular structure, making it difficult to navigate the specific property space [...]"
>
> We thank the reviewer for this important nuance. We note that Section 5.2 evaluates **structural** conditioning (atom composition, substructure fingerprints); global scalar properties such as $\log P$ or orbital energies represent a different task that was not evaluated.
>
> Regarding whether the motif vocabulary makes such conditioning harder: many practically relevant global descriptors, including $\log P$ and SA score, are explicitly defined as additive functions of chemical fragments, making motif-level representations a natural vehicle for such signals. For non-decomposable properties (e.g., orbital energies), a fragment-to-property mapping must be learned; however, this challenge applies equally to atom-based models, which also lack any guaranteed smooth relationship between continuous atomic trajectories and quantum-mechanical observables. We therefore do not view this as a fundamental disadvantage of motif-based generation, and leave global scalar property conditioning as future work.
>
> We thank the reviewer again and hope these clarifications address all remaining concerns.
>
> [1]: Equivariant Neural Diffusion for Molecule Generation (Cornet et al., 2024)

---

> > ### Author Rebuttal · Reviewer_2gyL · 2026-04-03
> >
> > I thank the authors for addressing my concerns, and I have decided to maintain my score.

---

### Official Review · Reviewer_aazF · 2026-03-13

**Soundness:** 3
**Presentation:** 3
**Significance:** 3
**Originality:** 3
**Overall Recommendation:** 5
**Confidence:** 4

**Summary:**

This paper introduce a method for generating 3D molecular structures using rigid motifs. The method uses discrete flow matching to select the motif types from a predefined vocabulary mined from the training dataset, each associated with a canonical pose, and uses flow matching on the SE(3)-manifold to orient the motifs into position. Strong results are reported for standard unconditional 3d molecule generation tests, as well as conditional generation. Ablation studies show the effects of the choice of fragmentation.

**Compliance With Llm Reviewing Policy:**

Affirmed.

**Final Justification:**

The paper represents a step forwards in the difficult task of using machine learning methods to propose novel molecules, highlighting how rigid fragments may be used. The proposed method outperforms other ML-based generative models for molecules, and the proposed rigid fragment representation is is something that future work can build off of.

**Key Questions For Authors:**

1. How does the specific dataset relate to the motif fragmentation? The results from Figure 5 are interesting to me but it is not clear if this is specific to QMUGs or if we would expect similar results for GEOM-Drugs and QM9. Would a dataset of smaller molecules like QM9 be more easily covered by a smaller set of motifs?
2. Was the scheduling between discrete motifs and continuous SE(3) transformations investigated? It seems like the rotation of a motif is arbitrary if it isn't yet assigned a set of coordinates. For example, Bose et al. (2023) found inference annealing useful, and in a crystal generation context, Miller et al. (2024) found annealing useful only on atomic coordinates but not on crystal lattices.

Bose, A. J., Akhound-Sadegh, T., Huguet, G., Fatras, K., Rector-Brooks, J., Liu, C. H., ... & Tong, A. (2023). Se (3)-stochastic flow matching for protein backbone generation. arXiv preprint arXiv:2310.02391.

Miller, B. K., Chen, R. T., Sriram, A., & Wood, B. M. (2024). Flowmm: Generating materials with riemannian flow matching. arXiv preprint arXiv:2406.04713.

**Limitations:**

Yes

**Strengths And Weaknesses:**

### Strengths
- Using motifs is well motivated and well demonstrated in the 2D molecular graph generation case, and this work nicely brings this technique into the 3D domain. It takes existing tools from prior works (e.g. FoldFlow and discrete flow matching) to solve the challenges presented by dealing with molecular motifs in 3D space (e.g. degeneracy of frames and discrete symmetries of motifs).
- The impacts of the architectural changes are well-explored through ablation studies.
- This method shows strong results in both the unconditional and conditional settings.
- This paper raises interesting points for future works to build off of: namely, that modeling at the level of individual atomic coordinates isn't necessarily well-motivated for molecule design, and that larger motifs may be used instead, but at the expense of generalizability
### Weaknesses
- The motif vocabulary is linked to the dataset, but this aspect isn't explored at all. Ablations on fragmentation are conducted on the QMUGs and GEOMDrugs dataset separately, without any interpretation of the difference between these datasets or explanation of why the QMUGs dataset is introduced for only this study.
- More information on statistics of the motif vocabulary would be useful. For example: figures showing the most common motifs, and the least common motifs for a given threshold occurrence. Histograms showing the frequencies of motif vs their rank, or showing the distribution of sizes of motifs.
- More explanation of prior research into fragments and motifs, even if they are for 2d graphs, would be helpful. For example, what makes a motif chemically meaningful? How does this relate to synthesizeability? Some other relevant citations:
    - Park, J., & Shen, Y. (2024). Equivariant blurring diffusion for hierarchical molecular conformer generation. Advances in Neural Information Processing Systems, 37, 131645-131675.
    - Levy, D., & Rector-Brooks, J. (2023). Molecular fragment-based diffusion model for drug discovery. In ICLR 2023-Machine Learning for Drug Discovery workshop.
    - Lee, J., Kim, S., Moon, S., Kim, H., & Kim, W. Y. (2025). FragFM: Hierarchical Framework for Efficient Molecule Generation via Fragment-Level Discrete Flow Matching. arXiv preprint arXiv:2502.15805.
    - Gottipati, S. K., Sattarov, B., Niu, S., Pathak, Y., Wei, H., Liu, S., ... & Bengio, Y. (2020, November). Learning to navigate the synthetically accessible chemical space using reinforcement learning. In International conference on machine learning (pp. 3668-3679). PMLR.

### Minor points:
- It would be more clear to write the fragmentation occurrence threshold as "α = 0.1%" consistently, rather than just "α=0.1" and separately noting that $α$ is a percentage.

---

> ### Author Rebuttal · Authors · 2026-03-31
>
> We thank the reviewer for their thoughtful review of our work. In the following, we address the reviewer’s concerns and questions.
>
> > Q1.1: Dataset–fragmentation relationship
>
> We reproduced the motif-sampling analysis of Fig. 5 for GEOM-Drugs and observed qualitatively identical results: common motifs are sampled at close-to-true occurrence across all strategies, while uncommon motifs are oversampled under No Rings and progressively undersampled as $\alpha$ decreases under Planar Rings. This confirms the finding generalises beyond QMugs and is not dataset-specific. Figures (including this GEOM-Drugs analogue and vocabulary statistics; see below) are provided at an anonymous link: https://zenodo.org/records/19340845.
>
> > Q1.2: “Would a dataset of smaller molecules like QM9 be more easily covered by a smaller set of motifs?”
>
> Not necessarily. QM9, while storing small molecules, is a near-exhaustive enumeration of all stable molecules up to 9 heavy atoms [1, 2], yielding substantial topological and geometric diversity. Concretely, for QM9: No Rings gives $|V_m|=49$; Planar Rings 0.01% gives $|V_m|=621$, falling between $\alpha=0.1$% ($|V_m|=202$) and $\alpha=0.01$% ($|V_m|=867$) for GEOM-Drugs, a dataset of much larger molecules. Vocabulary size is thus not simply governed by molecule size.
>
> > Q2: Scheduling between discrete and SE(3) flows
>
> To examine this interplay, we performed additional experiments. Unlike inference annealing in [3] (a numerical trick for boundary handling, which we do adopt, as noted in App. A.2) and [4], explicit modality scheduling requires decoupling components during *training*. We follow [5, 6] and train two separate temporally non-linear interpolants $\alpha_t = 1 - \cos^2 (\frac{\pi}{2}t^\nu)$, testing different $\nu$ for the discrete and SE(3) components. Larger $\nu$ means slower transition from prior to data, i.e., the state remains noisy/masked longer. We focus on two configurations on GEOM-Drugs (100 steps):
>
>
> | Schedule	| Atom Stability, \% | 	Validity $V$, \% | 	Connectivity $C$, \% | 	$V \times C$, \% |
> | :--- | :--- | :--- | :--- | :--- |
> | (1) $\nu_{\text{motif}} = 1, \nu_{\mathrm{SE}(3)} = 2$ | $97.3 \pm 0.0$	| $86.2 \pm 0.3$ | 	$72.4 \pm 0.9$	| $62.4 \pm 1.0$ |
> | (2) $ \nu_{\text{motif}} = 2, \nu_{\mathrm{SE}(3)} = 1$	| $92.7 \pm 0.0$ | $77.9 \pm 0.3$ | $96.0 \pm 0.2$	| $74.8 \pm 0.4$ |
> | MotiFlow (no schedule)	| $95.0 \pm 0.0$	| $87.2 \pm 0.3$	| $93.1 \pm 0.2$	| $81.2 \pm 0.3$ |
>
> In (1), motif identities are established early, yielding stable local chemistry, but the SE(3) flow struggles to merge fixed, disparate rigid bodies into a single molecule (low connectivity). Conversely, in (2), the SE(3) flow successfully connects abstract masked shapes early (high connectivity), but the categorical flow is then forced to assign discrete classes to a pre-existing 3D layout, causing valency mismatches (lower stability/validity). Our concurrent evolution in MotiFlow avoids both failure modes via self-conditioning: motif geometry is guided by the current estimate of discrete identities at each step, validated in architectural ablations (App. A.4.4, Table 12) and consistent with findings in [7]. We will include this schedule analysis in the revised Appendix.
>
> > Why is QMugs used only for ablations?
>
> To our knowledge, QMugs has not been used to benchmark generative models, so we lacked direct baselines for Sections 5.1–5.2. We introduce it in the ablation study to encourage the community to adopt this large-scale, chemically meaningful dataset and offer our results as a reproducible first baseline for future work.
>
> > Vocabulary statistics
>
> We agree with the suggestion and provide histograms of motif frequencies, size distributions, and motif examples at the Zenodo link above (see Q1.1).
>
> > Prior work on fragments and motifs
>
> We agree with the reviewer and appreciate the suggested literature. We will update the manuscript to expand on this discussion, ensuring we reference the suggested papers.
>
> > Minor
>
> We agree on the threshold notation and will adopt it in the camera-ready version.
>
> We appreciate the thorough work of the reviewer and hope to have addressed all their questions and concerns. Should any further questions arise, we welcome further discussion.
>
> [1]: Quantum chemistry structures and properties of 134 kilo molecules (Ramakrishnan et al., 2014)
>
> [2]: Top-N: Equivariant set and graph generation without exchangeability (Vignac et al., 2022)
>
> [3]: SE(3)-Stochastic Flow Matching for Protein Backbone Generation (Bose et al., 2024)
>
> [4]: FlowMM: Generating Materials with Riemannian Flow Matching (Miller et al., 2024)
>
> [5]: MiDi: Mixed Graph and 3D Denoising Diffusion for Molecule Generation (Vignac et al., 2023)
>
> [6]: Mixed Continuous and Categorical Flow Matching for 3D De Novo Molecule Generation (Dunn & Koes, 2024)
>
> [7]: Harmonic Self-Conditioned Flow Matching for Multi-Ligand Docking and Binding Site Design (Stärk et al., 2024)

---

> > ### Author Rebuttal · Reviewer_aazF · 2026-03-31
> >
> > I thank the authors for fully addressing each of the points that I have raised, and have increased my score.
> >
> > EDIT: I now have an additional question: With the set of hyperparameters originally presented, the models performs poorly at generating unique molecules for QM9, compared to other methods. As you demonstrated in your reply to reviewer fQUj, this represents a trade-off, and can be completely fixed by modifying the temperature schedule, at the expense of lowering validity. Does this tradeoff exist on the GEOM dataset? Can you confirm the percentage of unique and of novel molecules for the results for the GEOM dataset? If, unlike other methods, MotiFlow does not generate entirely unique molecules, can the sampling temperature schedule be changed so that it does, and what is the resulting stability, validity, and connectedness of the resulting molecules? Apologies for the late request.

---

> > > ### Author Response · Authors · 2026-04-08
> > >
> > > We sincerely thank the reviewer for raising their score and for the follow-up question regarding uniqueness and novelty on the GEOM-Drugs dataset.
> > >
> > > On the generated molecules from the GEOM-Drugs unconditional generation experiment (Section 5.1), MotiFlow achieves:
> > >
> > > * Uniqueness $U$, %: $99.92 \pm 0.01$
> > > * Novelty, %: $99.96 \pm 0.02$ (percentage of sampled molecules not present in the training set).
> > >
> > > These results confirm that the stability-uniqueness trade-off observed on the QM9 dataset of much smaller molecules does not exist for medium- and large-scale molecules of GEOM-Drugs.
> > >
> > > As noted in Section 5.1 (paragraph "Metrics"), we initially omitted these metrics from the main table because uniqueness on GEOM-Drugs is $\approx 100$% across methods. We appreciate the opportunity to provide the exact numbers, explicitly confirming to readers that this is the case for MotiFlow as well; we will add these results to the appendix of the final version of the paper.
> > >
> > > We thank the reviewer once again for their careful review and constructive feedback, which improved our manuscript.

---

### Decision · Program_Chairs · 2026-04-30

**Decision:**

Reject

**Comment:**

Dear authors, thank you for your submission and for engaging in detailed discussions with all reviewers. Since some reviewers were unavailable to quickly reply when the authors were allowed to respond, we hope that the comments below will help improve your work.

The stability score is defined in A.3.2 as the percentage of molecules that are charge neutral. This is indeed a natural heuristic for molecule generation and prediction, because a non-neutral molecule might exist under special ambient conditions, which requires a proper experimental confirmation in a wet lab. For this basic charge neutrality (called stability), the proposed MotiFlow achieves 92.6%, less than the QM9 baseline of 95.2% in Table 1.

On the second dataset GEOM, there is no reported stability at the molecular level, but the paper reported the VxC score (validity by RDKit and connectivity to guarantee a single molecule after adding chemical bonds under standard conditions) of 81.2% on MotiFlow, which is significantly less than the baseline of 99% on GEOM.

For canonical frames and comparisons, the paper mentioned the "rotation matrix, computed via the Kabsch algorithm". This algorithm (https://en.wikipedia.org/wiki/Kabsch_algorithm) essentially uses ordered points. If atoms are not ordered, the same approach leads to exponentially many permutations. Even if we take chemical elements into account, the simple molecule of benzene C6H6 has (6!)^2>500,000 possible permutations of atoms of the same chemical element.

Though the authors included "novelty" as "percentage of sampled molecules not present in the training set" in some responses, this word did not appear in the paper. The previous paragraph shows why quantifying the novelty of 3D molecules was highly non-trivial and was solved in general position by a distribution of pairwise distances at https://www.sciencedirect.com/science/article/pii/S0196885803001015 and for all unordered clouds by Lipschitz continuous invariants and metrics in polynomial time at https://openaccess.thecvf.com/content/CVPR2023/html/Widdowson_Recognizing_Rigid_Patterns_of_Unlabeled_Point_Clouds_by_Complete_and_CVPR_2023_paper.html

Uniqueness was defined in lines 734-735 as "the percentage of valid samples that possess a unique SMILES string". Unfortunately, the SMILES string depends on many choices and is unsuitable for uniqueness. Here is the quote from https://en.wikipedia.org/wiki/Simplified_Molecular_Input_Line_Entry_System: "The resultant SMILES form depends on the choices: of the bonds chosen to break cycles, of the starting atom used for the depth-first traversal, and of the order in which branches are listed when encountered."

Without proper comparisons of 3D molecules, one can easily generate millions of differently looking molecules through atom reordering and perturbations. In a similar case, the Nature correction at https://static-content.springer.com/esm/art%3A10.1038%2Fs41586-025-09992-y/MediaObjects/41586_2025_9992_MOESM1_ESM.pdf crossed out almost all the words "novelty" and "discovery" because nothing novel was discovered.

Potentially, the most impactful contribution is the exposure of disconnected entries in the training datasets QM9 and GEOM, while MotiFlow reported even lower percentages (92.6% < 95.2% on QM9 in Table 1 and 81.2%<99% on GEOM drugs in Table 2). Since the charge neutrality and connectivity of generated molecules should be easy to verify, the authors are encouraged to report all failures in their update. This exposure of past mistakes should lead to corrections of the QM9 and GEOM papers, similar to the Nature 2023 paper above.

Reviewer fQUj summarized the concerns as "Limited performance on small molecules".